# Long-read metagenomics of soil communities reveals phylum-specific secondary metabolite dynamics

Marc W. Van Goethem[1], Andrew R. Osborn[2], Benjamin P. Bowen [1], Peter F. Andeer[1], Tami L. Swenson[1,6], Alicia Clum[2], Robert Riley[2], Guifen He[2], Maxim Koriabine[2], Laura Sandor[2], Mi Yan[2], Chris G. Daum [2], Yuko Yoshinaga[2], Thulani P. Makhalanyane [3], Ferran Garcia-Pichel [4,5], Axel Visel [2], Len A. Pennacchio [2], Ronan C. O'Malley [1,2] & Trent R. Northen [1,2 ✉]

Microbial biosynthetic gene clusters (BGCs) encoding secondary metabolites are thought to impact a plethora of biologically mediated environmental processes, yet their discovery and functional characterization in natural microbiomes remains challenging. Here we describe deep long-read sequencing and assembly of metagenomes from biological soil crusts, a group of soil communities that are rich in BGCs. Taking advantage of the unusually long assemblies produced by this approach, we recovered nearly 3,000 BGCs for analysis, including 712 full-length BGCs. Functional exploration through metatranscriptome analysis of a 3-day wetting experiment uncovered phylum-specific BGC expression upon activation from dormancy, elucidating distinct roles and complex phylogenetic and temporal dynamics in wetting processes. For example, a pronounced increase in BGC transcription occurs at night primarily in cyanobacteria, implicating BGCs in nutrient scavenging roles and niche competition. Taken together, our results demonstrate that long-read metagenomic sequencing combined with metatranscriptomic analysis provides a direct view into the functional dynamics of BGCs in environmental processes and suggests a central role of secondary metabolites in maintaining phylogenetically conserved niches within biocrusts.

[1] Environmental Genomics and Systems Biology, Lawrence Berkeley National Laboratory, 1 Cyclotron Rd, Berkeley, CA 94720, USA. [2] DOE Joint Genome Institute, Lawrence Berkeley National Laboratory, 1 Cyclotron Rd, Berkeley, CA 94720, USA. [3] Centre for Microbial Ecology and Genomics, Department of Biochemistry, Genomics and Microbiology, University of Pretoria, Lynnwood Rd, Hatfield, Pretoria 0028, South Africa. [4] Center for Fundamental and Applied Microbiomics, Biodesign Institute, Arizona State University, Tempe, AZ, USA. [5] School of Life Sciences, Arizona State University, Tempe, AZ, USA. [6] Present address: Labcorp Drug Development, Covance, Madison, WI, USA. ✉email: TRNorthen@lbl.gov

A fundamental challenge in understanding the ecological functions of secondary metabolites (also known as specialized metabolites or natural products) is that most biosynthetic gene clusters (BGCs) are harbored by uncultivated microbes and require specific native contexts for activation[1]. The majority of BGCs, also referred to as microbial gene clusters, encoding secondary metabolites are not usually expressed under standard cultivation conditions in the laboratory[2] and their products have therefore been termed "secondary" metabolites. A universal feature of BGCs is their modular, co-localized gene architecture[3] and large size, frequently spanning tens of thousands of base pairs. Bacterial secondary metabolites play critical ecological roles in mediating communication, antagonistic interactions, nutrient scavenging, and have historically been a primary source for antibiotic drug development;[4] in fact more than half of registered drugs are based on natural secondary metabolites[5]. Additionally, secondary metabolites have applications in agriculture[6], biomaterials[7], biofuels[8], and cosmetics[9].

Previous work has demonstrated the potential for deep shotgun metagenomic sequencing to directly characterize BGCs from environmental samples[10,11], but the assembly of full-length BGCs from short reads is associated with major limitations[12]. Notably, BGCs are almost always part of the flexible, rather than core, genome, which can assemble poorly using short read metagenomes[13]. Alternative techniques include the use of clone libraries[1] or innovative sequence-based analyses[14,15] including the reconstruction of uncultivated microbes as metagenome-assembled genomes (MAGs; reviewed in[16]). Although these approaches typically give access to both dominant and rare members of the community, many contigs will not be binned into a genome[17]. Moreover, long-read sequencing circumvents the requirement of generating MAGs in some cases, as large genome segments are captured directly through sequencing and assembly, which could favor low-abundance species.

We also know remarkably little about the transcription of BGCs in nature or how the environment regulates their production[18] especially in soils. This information is critical in understanding how often secondary metabolites are produced in natural communities. Biological soil crusts (biocrusts) are the world's most extensive biofilms and together cover up to 12% of total soil surface area[19]. Initial studies have suggested that they are rich in secondary metabolites[20]. Cyanobacteria dominate biocrust communities, specifically Microcoleus spp. that drive biocrust establishment by stabilizing the soil surface, both preventing erosion and improving soil fertility through the release of photosynthate[21,22]. In contrast to many other types of soil environments, biocrusts are easily transferable to the laboratory, which allows for controlled interrogation of relevant environmental processes such as wetting dynamics. In native environments, rain events suspend microbial dormancy in biocrust and cause dramatic shifts to community structure and both primary and secondary metabolite release[23]. The secondary metabolites produced by microbes upon wetting are known to include antimicrobial compounds thought to provide a selective advantage[24], yet the majority of secondary metabolites encoded in the genomes of biocrust community members remain unidentified[25]. Cyanobacteria are known secondary metabolite producers[3,26] but most studies have focused on aquatic cyanobacteria, leaving the secondary metabolites of terrestrial cyanobacteria largely underexplored[27,28].

We combined long- and short-read metagenomic sequencing to produce large assemblies that enabled BGC discovery. We then mapped time-series metatranscriptomes to gain insight into the environmental cues governing BGC expression in wetted biocrusts. Our results showed that thousands of gene clusters could be extracted from assembled long-read metagenomes which gave insight into the secondary metabolism of uncultivated microbial taxa. Coupling these results to metatranscriptomics indicated that most BGCs were transcribed after a simulated rain event, and that cyanobacteria dominated secondary metabolism throughout the experiment.

## Results and discussion

**Long-read sequencing permits access to biosynthetic gene clusters.** Biocrust samples were collected from Moab, UT, USA (Fig. 1a), and transported to the JGI in petri dishes that maintain the physical structure of the crust. We then extracted and prepared high-molecular-weight DNA from an intact biocrust sample to be sequenced on both long- and short-read metagenomic sequencing platforms (Supplementary Fig. 1). In total, we sequenced eight SMRT cells from three libraries yielding 156.3 Gb from 36.7 million reads, where half of all sequenced bases were contained in reads of 5 kb or longer, while the longest read was 167 kb. The average read length was 3,084 bp while the mean $N50$ value was 4070 bp. Both statistics were augmented by the Sequel II library which comprised 108 Gb of sequence in just 19.1 million reads.

The two short-read Illumina libraries provided an additional 20 Gb of sequence (Supplementary Data 1). To obtain an initial phylogenetic profile of the communities under investigation we performed full-length 16S rRNA gene analysis using exact sequence variants (ESVs) which showed that Cyanobacteria, and particularly Microcoleus vaginatus, were dominant biocrust community members in most samples, with major representations of Actinobacteria and Alphaproteobacteria (Fig. 1b) which is generally consistent with the known community composition of these biocrusts[29]. Overall, the biocrust are less complex than other desert soil communities[30] yet are notably richer in cyanobacteria. We found very few sequences for eukaryotes after seeking 18S rRNA genes, range 100–1000 sequences per metagenome, and acknowledge that more attention should be given to the diverse members of kingdom in future research as fungi, rotifers, algae and mosses can constitute important components of the biocrust microbiome[31,32]. While this work was focused on BGCs for bacteria, we saw a small number of eukaryotic reads (>1% of all sequences, although no BGCs) and this would be an interesting investigation for a future analysis of this dataset, given their importance in late-stage biocrust[33].

To access biosynthetic gene clusters, we individually assembled the biocrust metagenomes into contiguous sequences (contigs). Using both Canu[34] and metaFlye[35] we assembled the long-read (n = 8 SMRT cells, 74,953 contigs, N50 = 18.2 kb) into assemblies that totaled 781 Mb in size, with half of the sequence present in contigs longer than 20 kb. The longest contig was more than 753 kb in length assembled from the largest long-read metagenome (Supplementary Data 2). The two short-read Illumina libraries assembled into ~8 million contigs (3.7 Gb, N50 = 1 kb).

We also co-assembled the metagenomes to access even more BGC diversity than was permissible from the individual assemblies. Firstly, we co-assembled the five largest long-read metagenomes (Sequel and Sequel II metagenomes) which yielded 1.4 Gb of assembled sequence (Supplementary Data 2) with the longest contig exceeding 1.3 Mb in length (N50 = 36 kb). We omitted the RS II metagenomes from co-assembly since their sizes are small compared to the Sequel counterparts. This co-assembly was as large as our hybrid co-assembly of two short-read Illumina libraries and the four Sequel long-read libraries produced with metaSPAdes[36] (1.7 Gb, N50 = 2.3 kb). Putative misassemblies identified through MetaQUAST were identified and removed[37]. Overall, the long-read assemblies and co-assemblies produced the largest number of long contigs (>50 kb) and were thus most suited

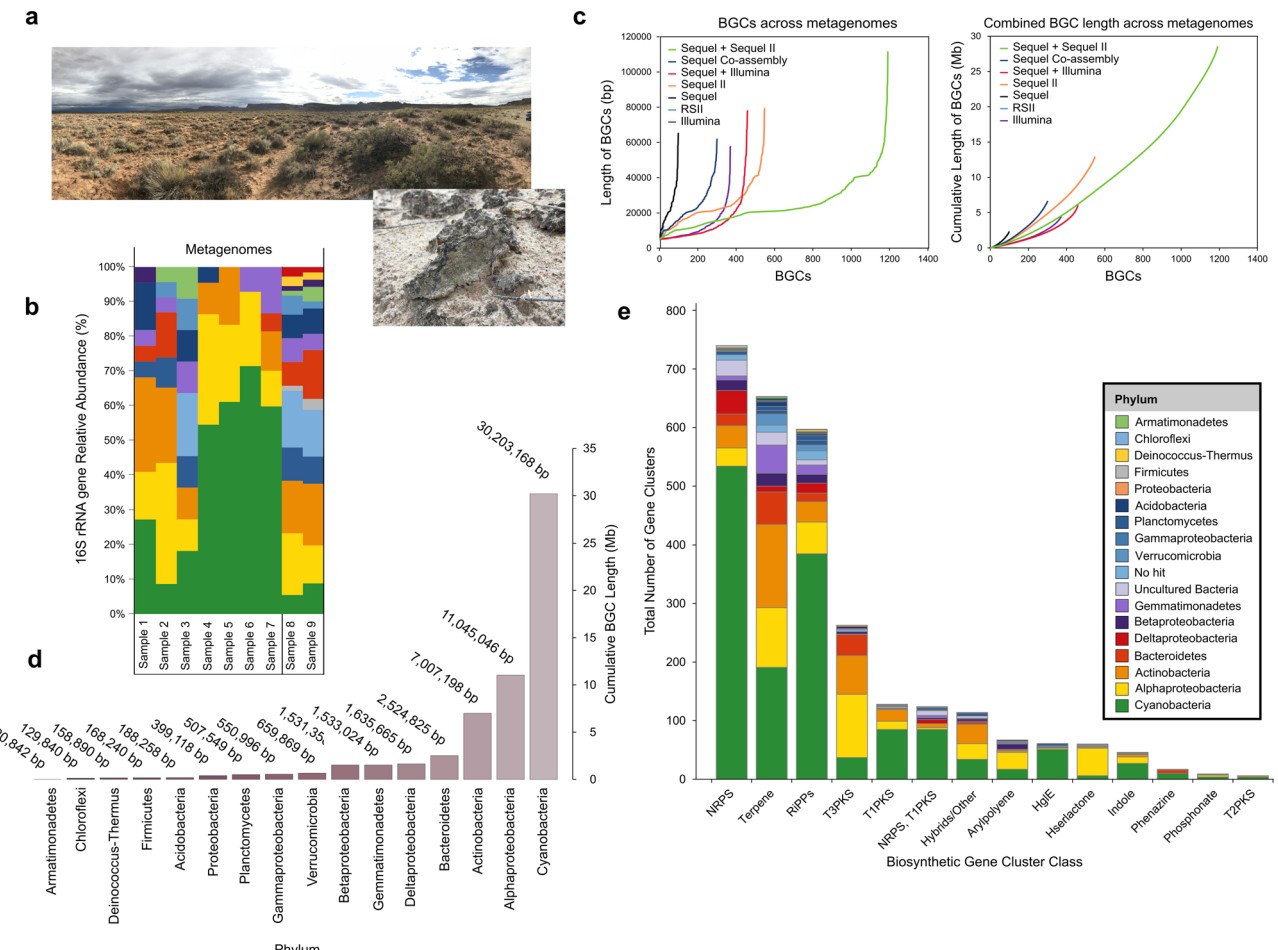

**Fig. 1 Secondary metabolism of biocrust. a** Photos of Biocrusts taken near Canyonlands National Park (Moab, UT) with the biological soil crust inlay showing the characteristic green coloration. Photo credit Trent R. Northen. **b** Taxonomic composition of the metagenomes based on exact sequence variants (ESVs) of 16 S rRNA genes across sequencing platforms. Relative abundances were calculated after assigning taxonomy against the SILVA reference database. **c** Left panel shows the number of Biosynthetic Gene Clusters (BGCs) recovered from each assembly, arranged from shortest to longest. Right panel shows the cumulative length of BGCs recovered from each metagenome in Megabases (Mb). **d** Taxonomic distribution of BGCs in megabase pairs (Mb) at the phylum or class level. **e** BGCs longer than 5 kb from each major class of secondary metabolism, colored by putative phylum-level assignments.

for the investigation of full-length biosynthetic gene clusters. Together they gave unprecedented access to the BGCs encoded by uncultivated microbes including 1191 BGCs from the long-read co-assembled metagenome.

Overall, the long-read metagenomes, and particularly their co-assemblies, offered substantially deeper insight into biocrust secondary metabolism than was possible through short-read sequencing and assembly (Supplementary Fig. 2). For example, the Sequel II assembly had 548 BGCs including 174 full-length BGCs (i.e., the BGC was not truncated on either contig edge), while the short-read assemblies had 359 BGCs between them yet only 9 full-length BGCs. In total, that is, including all assemblies, we predict that 712 BGCs are full-length clusters.

To put our assemblies in context of existing sequence technology, our long-read co-assembly is ~90% of the very large short-read Iowa prairie metagenomes at 1.5 billion bp[38], but 1473% and 1688% of the biogas and Lake Biwa long-read co-assemblies, respectively[39,40]. Despite their smaller size compared to short-read metagenome assemblies, we get a higher percentage of long contigs from long-read metagenomes which is paramount for BGC detection.

The single largest BGC was identified in the large co-assembly and was putatively assigned to the genus *Nostoc*. It encodes a previously undescribed hybrid transAT-polyketide synthase-nonribosomal peptide synthetase of 111 kb length, harboring six core biosynthetic genes and eight additional biosynthetic genes. Manual inspection suggests it is full-length, making it one of the longest BGCs to be identified directly from a soil metagenome (Supplementary Fig. 3). The co-assembly of multiple long-read metagenomes offered deeper access to the spectrum of BGCs while the diversity of these clusters found here suggests that much secondary metabolic potential remains unrealized in current databases. Moreover, the use of long-read sequencing is central to finding full-length gene clusters, an issue that precluded the use of short-read metagenomics previously.

**Thousands of gene clusters recovered from biocrust metagenomes.** We performed gene cluster identification and annotation for secondary metabolites[41] using all the *de novo* metagenome assemblies owing to their high contiguity in assembly and high proportion of contigs longer than 5 kb ($n = 141,762$ total contigs; Fig. 1c). This approach recovered 2988 BGCs predicted to produce secondary metabolites from uncultivated biocrust microbes across all metagenome assemblies. These span all major secondary metabolite classes with terpenes, ribosomally synthesized and post-translationally modified peptides and non-ribosomal peptide

**Table 1 The number of biosynthetic gene clusters recovered from each assembly or co-assembly are shown with details regarding full-length BGCs, BGCs sequenced previously and which assemblies contributed to the co-assemblies.**

| Co-assembly/Assembly | Number of BGCs | Full-length BGCs | No. sequenced Previously | Sequencing Platforms |
|---|---|---|---|---|
| Flye_co-assembly | 300 | 67 | 28 | PacBio Sequel ($n = 4$) |
| Combined_Sequel_Illumina | 459 | 31 | 42 | PacBio Sequel ($n = 4$) & Illumina HiSeq 2500 ($n = 2$) |
| Ultimate_Sequel+SequelII | 1191 | 419 | 21 | PacBio Sequel ($n = 4$) & PacBio Sequel II ($n = 1$) |
| pbio-1768.15750 | 16 | 0 | 3 | PacBio RS II |
| pbio-1772.15782 | 5 | 0 | 1 | PacBio RS II |
| pbio-1768.15751 | 0 | 0 | 0 | PacBio RS II |
| m54017_180413_173154 | 24 | 3 | 8 | PacBio Sequel |
| m54017_180414_134655 | 30 | 2 | 7 | PacBio Sequel |
| m54017_180414_220614 | 0 | 0 | 0 | PacBio Sequel |
| m54017_180417_205359 | 44 | 7 | 5 | PacBio Sequel |
| pbio-2210.20021 | 548 | 174 | 9 | PacBio Sequel II |
| 11774.4.218925 | 139 | 3 | 19 | Illumina HiSeq 2500 |
| 12041.5.235284 | 232 | 6 | 32 | Illumina HiSeq 2500 |

| Co-assembly/Assembly | Metagenomes used for co-assembly/assembly |
|---|---|
| Flye_co-assembly | m54017_180413_173154, m54017_180414_134655, m54017_180414_220614, m54017_180417_205359 |
| Combined_Sequel_Illumina | m54017_180413_173154, m54017_180414_134655, m54017_180414_220614, m54017_180417_205359, 11774.4.218925, 12041.5.235284 |
| Ultimate_Sequel+SequelII | m54017_180413_173154, m54017_180414_134655, m54017_180414_220614, m54017_180417_205359, pbio-2210.20021 |
| pbio-1768.15750 | pbio-1768.15750 |
| pbio-1772.15782 | pbio-1772.15782 |
| pbio-1768.15751 | pbio-1768.15751 |
| m54017_180413_173154 | m54017_180413_173154 |
| m54017_180414_134655 | m54017_180414_134655 |
| m54017_180414_220614 | m54017_180414_220614 |
| m54017_180417_205359 | m54017_180417_205359 |
| pbio-2210.20021 | pbio-2210.20021 |
| 11774.4.218925 | 11774.4.218925 |
| 12041.5.235284 | 12041.5.235284 |

synthetases particularly well represented. Cyanobacteria were rich in non-ribosomal peptide synthetases and Type 1 polyketide synthases and harbored the most BGCs overall encoding some 1470 BGCs (Fig. 1d; Supplementary Data 3). Four hundred and twenty of these non-redundant BGCs could be assigned to the genus *Microcoleus*—the pioneer microbial guild of biocrust[42].

Next, we determined whether previous sequencing efforts had captured the BGCs previously by making queries to the entire NCBI nt sequence database (accessed December 6, 2019[43]). Using thresholds of 75% sequence identity over 80% of the sequence length[44] we identified 175 BGCs that had been sequenced previously. Thus ~94% of BGCs had not been sequenced before (Table 1). This reaffirms that biocrust are a rich source of BGCs relative to either aquatic systems such as Lake Biwa or biogas reactors which are simpler systems compared to soil. Moreover, these results underscore the potential for long-read metagenomic sequencing in BGCs discovery. We recognize that there is substantial opportunity to do more in-depth analyses of the diversity of BGCs recovered and feel that this presents a promising future research direction.

Of the known clusters, 143 belonged to Cyanobacteria including the late-branching genera *Microcoleus*, *Nostoc*, and *Oscillatoria*. BGCs identified from non-cyanobacterial contigs also had interesting elements. For example, *Planctomycetes* were rich in acyl-amino acids, while *Alphaproteobacteria* had unusually high numbers of the dipeptide N-acetylglutaminylglutamine amides as well as N-acyl-homoserine lactones that may be involved in quorum sensing[45]. Moreover, many terpenes and Type 3 polyketide synthases belonged to the dominant heterotrophic phyla *Alphaproteobacteria* and *Actinobacteria* (Fig. 1e). We also found 17 phenazines in our dataset, most of which belonged to cyanobacteria, and some of which may have functions in redox balance during anoxia[46], which frequently occurs in biocrust after intense rain events. Next we sought out to resolve the environmental stimuli that induce BGC transcription after wetting events in order to link BGCs to their role in biocrust ecology. Here we opted to analyze all BGCs simultaneously in an effort to unearth variations between their transcriptional profiles that could emerge following wetting, for example.

**Constitutive transcription of secondary metabolite gene clusters.** Desert biocrust communities are sensitive to rain events, as revealed by dramatic changes in microbial community structure[29] and core gene expression by RNA microarray[23,47]. To identify secondary metabolite BGCs involved in these dynamics, we mapped 13 biocrust metatranscriptomes to our metagenome assemblies. The metatranscriptomes are from a simulated rain event in the laboratory using intact biocrust from the same site (Moab, UT, USA)[48]. They capture microbial transcription following a wetting event for three diurnal cycles at a resolution of ten individual timepoints. Like the metagenomic data, 16S rRNA transcript analysis using ESVs from the metatranscriptomic datasets revealed an abundance of transcripts from *Cyanobacteria*, and especially *Microcoleus vaginatus* at all timepoints (Fig. 2a). We observed a dramatic increase in 16S rRNA transcript copy numbers across all taxa 15 min and 1 h after wetting possibly indicating increased microbial growth on substrates releaseds during cell membrane permeabilization after wetting[49] or simply ribosome synthesis as microbes emerge from dormancy (Supplementary Data 9).

The metatranscriptomic data comprised 137 Gb of high-quality sequence in 919 million transcripts from 13 samples

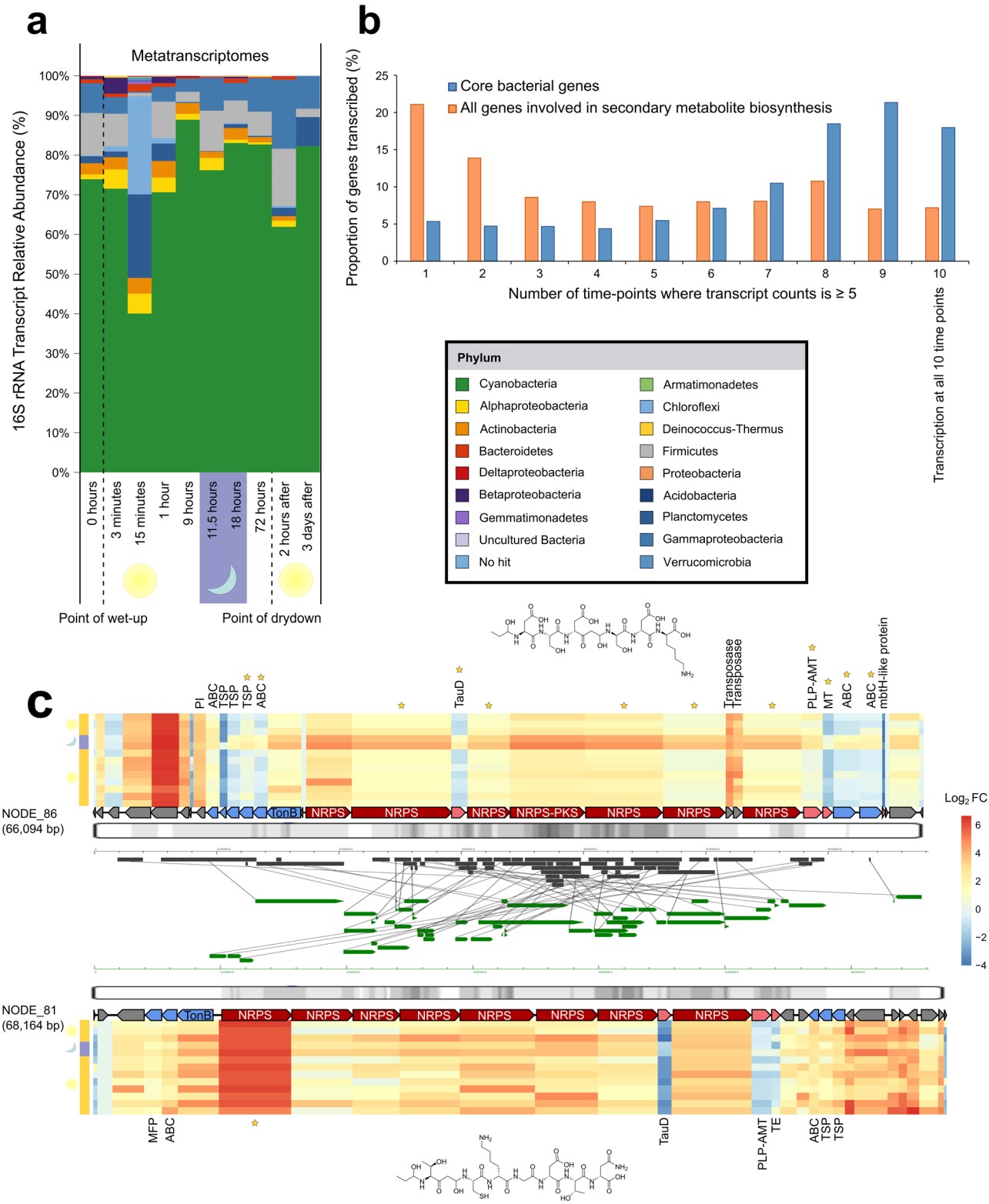

(Supplementary Data 1, 2 and 4). To calculate secondary metabolite gene transcription after wetting we mapped the individual read transcripts to each contig containing a BGC using BBMap[50] (*ref* = assembled metagenome, *in* = filtered metatranscriptomic sequences, *outm* = transcripts mapped to contigs.sam) which leveraged our long contigs to profile transcription for almost 3000 secondary metabolite gene clusters. Remarkably, we found that 395 biosynthetic genes from 240

BGCs were transcribed at all timepoints (using a threshold of at least five mapped transcripts per gene within a cluster at a single time point which excluded low levels of read mapping), which represent some 6% of all secondary metabolic genes in our dataset (Fig. 2b). Most of the constitutively transcribed biosynthetic genes within BGCs were AMP-binding domains, condensation domains, PCP domains and phytoene synthases, all of which have central roles in the biosynthesis of specialized metabolites.

**Fig. 2 Transcription of secondary metabolites. a** Taxonomic composition of the metagenomes based on exact sequence variants (ESVs) of 16S rRNA transcripts during a soil wetting experiment. Relative abundances were calculated after assigning taxonomy against the SILVA reference database. **b** Core bacterial gene transcription ($n = 46$ genes including DNA-binding proteins, Large and Small subunit ribosomal proteins) shown in blue compared to secondary metabolite gene transcription (orange). Genes transcribed at all 10 timepoints (rightmost point) are thought to experience constitutive expression. The y-axis indicates the proportion of genes transcribed at each timepoint category. **c** Two putative rearranged siderophore-producing gene clusters found in the co-assembled metagenomes that showed homology are aligned. The heatmaps indicate the transcription of each gene in the BGC based on statistical tests against the control sample; 0 h at the point of wetting. Heatmap columns are scaled to the size of the mapped gene, and row order ($n = 13$ rows) indicates progression across the experiment from 0 h (bottom row) to 3 days after wetting (top row). The left-hand column shows day (yellow) and night (purple) categories. The predicted chemical structure of NODE_86 is shown above while the structure of NODE_81 is below the contig. Transcriptional profiles of gene clusters with differentially expressed genes at night are shown with stars.

*Cyanobacteria*, and to a lesser extent *Actinobacteria*, encoded most of these biosynthetic genes while our results further indicate that non-ribosomal peptide synthetases and polyketide synthases gene clusters were the most prominent class of secondary metabolites that were constitutively transcribed. In some cases we observed that multiple consecutive biosynthetic genes within the same BGC were constitutively transcribed. On average we found that at least two biosynthetic genes on the contig were constitutively transcribed while single biosynthetic genes within a BGC experiencing constitutive transcription were rare (Supplementary Data 10).

Our results show stark contrast to previous observations that BGC expression in the laboratory is low wherein most secondary metabolites are not transcribed[2]. Their constitutive expression supports the notion that secondary metabolites may play critical (and possibly essential) roles in communication or niche occupancy in these ecosystems. Given the relatively high biosynthetic cost of synthesizing secondary metabolites vs. primary metabolites[51] this suggests that these compounds provide fitness benefits to their hosts across the wetting event. Furthermore, we recovered 88 BGCs from our assembled metatranscriptomes, 59 of which were transcribed 11.5 h after wetting. Most BGCs found in this dataset were non-ribosomal peptide synthetases while cyanobacteria were the most common source of these full-transcribed BGCs (Supplementary Data 4, Supplementary Note 1).

Next, we investigated how the observed constitutive expression of secondary metabolic genes compared to the transcription of all other genes, i.e., those not involved in secondary metabolism. Of the 966,111 "non-secondary" genes detected across all contigs, i.e., those with or without BGCs, just 43,139 (some ~4.5%) were constitutively transcribed at all 10 time points (Supplementary Data 5). These mapping rates were not artefacts of gene length differences between primary genes and secondary metabolic gene lengths (Supplementary Fig. 4). We then focused on 46 core metabolic bacterial genes that we expected to have high constitutive expression, e.g., those encoding DNA-binding or ribosomal subunit proteins (Supplementary Data 4), and found that indeed many of these core genes were transcribed at eight or more time points, and ~18% that were constitutively transcribed (5 mapped transcripts at all 10 timepoints; Fig. 2b). This same analysis of secondary metabolic genes showed a more even distribution across the time points with 6% transcribed at all 10 time points (Fig. 2b). Although lower than for core bacterial genes, this represents a higher proportion of constitutive transcription for secondary metabolic genes than was anticipated.

While our results show low level constitutive transcription of many BGCs, the highest level of BGC transcriptional activity occurred at night, 11.5 h after the initial wetting event (Supplementary Fig. 5a). This enrichment in transcription was mostly underpinned by a surge in transcriptional activity by the *Cyanobacteria* (Supplementary Fig. 5b) which likely corresponds to gene induction at night when they are not photosynthetically

active[47]. Strikingly, 80% of cyanobacterial BGC transcription peaked at night. This included the significant transcription of two putative siderophore-producing BGCs (DESeq2: $P < 0.05$), while their observed rearrangements were presumably driven by transposases (Fig. 2c, Supplementary Fig. 9 and Supplementary Note 3). We also observed another peak of BGC transcriptional activity 72 h after wetting (during the day, and the point of dry down) which was due to the increased transcription of terpenes and Type3 polyketide synthases by abundant heterotrophic bacteria such as *Deltaproteobacteria* and *Actinobacteria* (Supplementary Figs. 6–8, Supplementary Data 6).

To further examine the phylogenetic conservation of BGC transcription among phyla we analyzed a subset of biosynthetic genes individually ($n = 12,470$ genes) using t-SNE visualization[52]. This analysis revealed some segregation of biosynthetic gene transcription by taxonomy including for the *Cyanobacteria* and a few other phyla (Fig. 3a). A two-sided Pearson pairwise correlation analysis revealed that *Bacteroidetes* were the phylum most strongly correlated with the cyanobacteria (Pearson's $R = 0.858$, adjusted $p < 0.004$, with a one-step Bonferroni correction, Supplementary Data 8). This is interesting because *Bacteroidetes* have recently been identified as one of the keystone members of the biocrust cyanosphere[53] and are also common host-associated organisms[54].

While Cyanobacteria exhibited the highest level of BGCs transcription at night, 11.5 h after wetting, most other bacteria (in this case almost exclusively heterotrophic guilds) showed maximal BGC transcription during the day (Supplementary Fig. 5). 82 BGCs that were maximally transcribed at 11.5 h included 70 cyanobacterial BGC, while 6 were assigned as *Bacteroidetes*.

Analysis of single gene expression is inherently more complicated to interpret against phylogenetic clusters due to the large number of comparisons. We compared the degrees to which biosynthetic gene clusters shared similar transcriptional profiles across phyla using a co-occurrence network based on the average Z-scores of each BGCs transcription ($n = 2988$). This analysis revealed clustering of secondary metabolite transcription of entire BGCs by taxonomy. Namely, the bacterial phyla had distinct temporal signatures of BGC transcription compared to each other over the course of 3 days (Fig. 3b). Cyanobacterial BGC expression was distinct from most bacterial groups in the biocrust based on Pearson pairwise correlations (Fig. 3c; Pearson adjusted $P < 0.05$). To our knowledge, this is the first such observation of phylum-level differences in microbial BGC transcription in manipulated natural communities. This may reflect conservation of life history traits especially niche competition strategies. For example cyanobacteria can grow heterotrophically on diverse dissolved organic components[55] and increased BGC expression may reflect increased competition with heterotrophs occurring at night. Thus, at night *Microcoleus* and other cyanobacteria may produce antibiotics, such as many bacteriocins that we detected here, to antagonize heterotrophs competing for dissolved organic

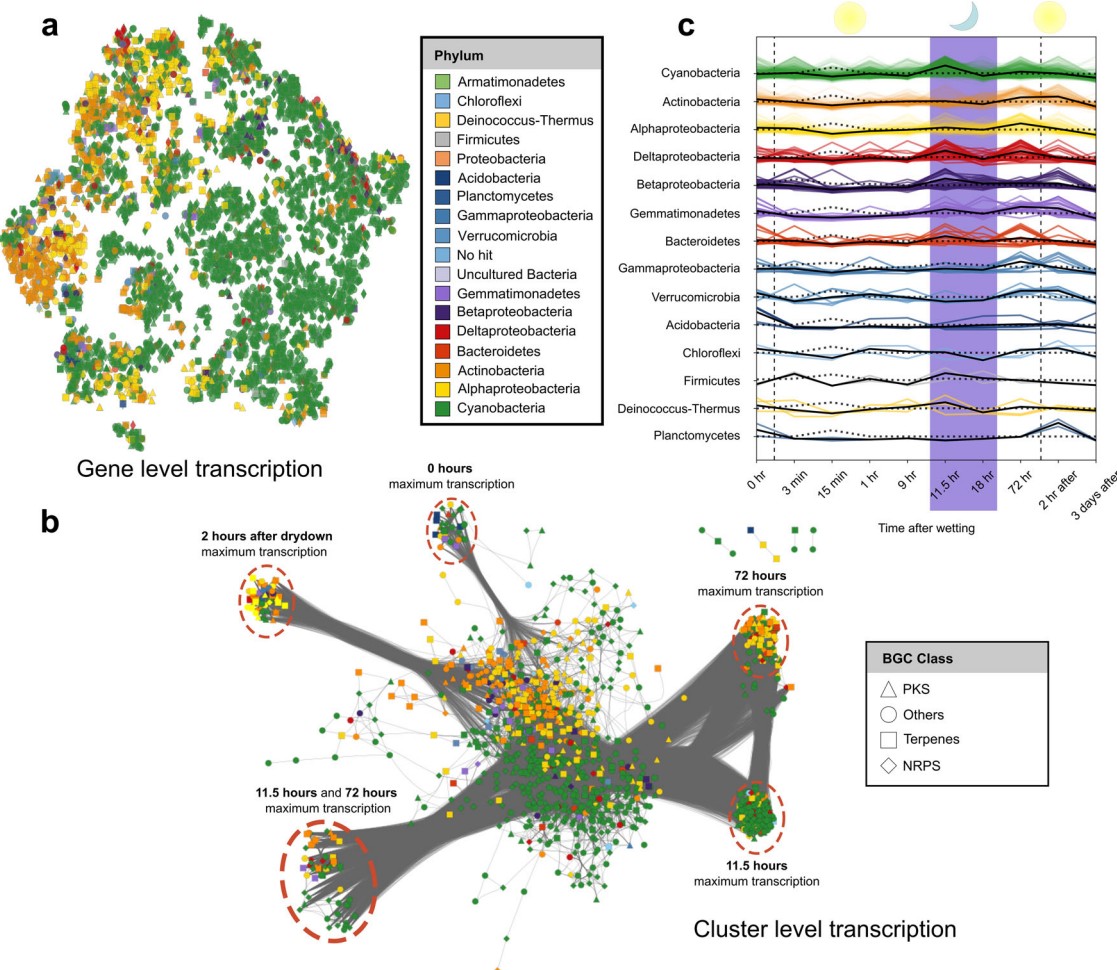

**Fig. 3 Phylum specific transcription of secondary metabolites. a** t-SNE visualization of every individual biosynthetic gene identified. The color of the points indicate the phylum assignment whilst shapes indicate the BGC class. **b** Co-occurrence network based on Pearson correlations ($r > 0.8$) among entire BGCs ($n = 2988$) based on average z-scores at each time point. Each node is a BGC within a contig that are colored by phylum and shaped by BGC type. Closely clustered nodes share similar transcriptional profiles. **c**, Line plot showing 16S rRNA transcript copy number over time shown by black, dotted lines. Average BGC transcription over time shown by the colored, solid lines. *Cyanobacteria* (green) show a unique night-time upregulation of secondary metabolism. Purple background indicates night-time transcription.

compounds[56], although we note that many other secondary metabolic functions may be performed at night including redox balancing or siderophore and pigment production.

Notwithstanding antagonism, the night-time expression of BGC products could facilitate electron and nutrient transport. Redox-active secondary metabolites are known to be produced by microbes under anoxic conditions[46]. For example, *Pseudomonas aeruginosa* enhances substrate-level phosphorylation during anoxia through the production of phenazines that facilitate electron transport 4[57]. Constitutive expression of the siderophore-producing gene clusters in cyanobacteria may reflect cation import strategies (notably iron scavenging) needed to support photosynthesis and other metabolic activities (Supplementary Note 2, Supplementary Data 7).

## Conclusion

In this study we show that long-read metagenomic sequencing is a powerful new tool for the examination of secondary metabolite gene clusters directly from complex environmental samples. Integration with metatranscriptomics revealed that ~6% of secondary metabolic genes were constitutively transcribed over 3 days—a higher percentage than many other genes. Thus, while conventionally unexpressed under laboratory conditions, our

results show that in situ BGCs appear to control important life history traits involved in maintaining microbial niches. BGC expression showed strong phylogenetic conservation where *Cyanobacteria*, unlike most other phyla, exhibited the highest levels of transcription at night. We speculate that this may reflect the switch from cyanobacteria serving as primary producers during the day to competing with heterotrophs for dissolved organics at night.

## Methods

**Biocrust sample collection and DNA isolation**. Biological soil crust (biocrust) was collected from Green Butte Site near Moab, UT, USA (38°42′54.1″N, 109°41′27.0″W) in 2014. Here we used a sterile petri dish to extract an intact biocrust after adding water to the surface to test for greening which indicated biological activity[23]. This field site is part of a long-term ecological research area of scientific interest aimed at exploring climatic changes in arid regions. We sampled early maturity biocrust (*Microcoleus*-dominated) by coring directly into the soil surface with a petri dish (6 cm² by 1 cm in depth). Samples were maintained in petri dishes in a dark desiccator in the laboratory until required for DNA isolation (~2 months). Previous studies have shown that biocrusts are viable for long-periods of times when stored under these conditions[58]. Metagenomic DNA was isolated using the MoBio Powersoil kit as per the manufacturer's instructions with a minor modification. We extracted DNA from 2 g of crust material by dividing the sample into four separate tubes (0.5 g in each tube). The nucleic acids from each tube were eluted in 50 µl of elution buffer and then pooled these into a final sample containing 200 µl of elution buffer and DNA.

**SMRT sequencing**. We sequenced three SMRT cells on the PacBio RS II Single Molecule, Real-Time (SMRT®) DNA Sequencing System (Pacific Biosciences, CA, USA) using two different library inserts: 10 kb AMPure PB library [$n = 2$] and a Low input 3 kb PB library [$n = 1$] using binding kit P6 v2 with 360-min and 120-min movies for the respective libraries. This biocrust sample was well-suited for long-read sequencing as it contained high-quality DNA without the need for additional steps to increase DNA fragment size. The same libraries were then sequenced on a PacBio Sequel System (Pacific Biosciences) using Sequel Binding Kit 2.1 with a combination of 600- and 1200-min movies. A third library was made using 10 kb AMPure PB approach with a Blue Pippin size cutoff of 4.5 kb. It was sequenced on PacBio Sequel II System (Pacific Biosciences) using 1.0 template prep kit and a 900-min movie.

To test how well-suited long-read metagenomes are for BGC recovery, we also made use of five publicly available PacBio SMRT metagenomes. The two datasets analyzed include a biogas reactor library sequenced on the PacBio RS II System with a 2 kb insert length[39], and four metagenomes obtained from Lake Biwa, Japan that were sequenced on a PacBio Sequel System with a 4 kb insertion length[40]. Raw sequence statistics for each metagenome is provided in Supplementary Data 1. We analyzed the sequencing effort of the metagenomes using Nonpareil v3.30[59] which relies on read redundancy. We performed a similar comparison using publicly-available long-read metagenomes which also yielded improvements in contig sizes and BGC yield from co-assembled datasets.

**Illumina sequencing**. Two unamplified 300 bp Illumina libraries were generated and sequenced 2 × 150 bp on the HiSeq-2500 1TB platform (Illumina).

**Taxonomy**. We extracted prokaryotic 16S rRNA genes using SortMeRNA 2.1b[60]. These 16 S rRNA sequences were then analyzed using DADA2[61] to identify exact sequence variants (ESVs) under default parameters with the exceptions of truncLen (150) and maxEE (1). The ESVs were then assigned taxonomy against the entire SILVA 16 S rRNA gene reference database[62]. The taxonomy of the identified gene clusters was inferred by BLAST queries[43] against the NCBI nr-database whereby hits were retained with E-values of less than $1 \times 10^{-10}$ and bit scores greater than 60.

**Assembly**. We performed read correction, trimming and assembly for the three RS II SMRT cells with Canu v1.8[34]. Here we included parameters suggested by the developers of Canu for PacBio metagenomes including an estimated mean genome size of 5 Mb (genomeSize = 5 m). We also changed the following parameters from their default values: corMinCoverage = 0, corOutCoverage = all, corMhapSensitivity = high, correctedErrorRate = 0.105, corMaxEvidenceCoverageLocal = 10 and corMaxEvidenceCoverageGlobal = 10.

The four larger Sequel metagenomes were assembled using metaFlye v2.4.2 under default settings with an estimated genome size of 5 Mb and the –meta option implemented for metagenomic sequence data[35]. All Illumina sequence data were quality trimmed prior to assembly using Prinseq-lite v0.20.4[63] with -min_qual_mean set to 20 and -ns_max_n set to 0 which eliminates low quality reads and ambiguous bases (internal N's). We assembled the two biocrust Illumina metagenomes with metaSPAdes v3.13.0[36] as recommended for paired-end short read length Illumina libraries[64]. We also co-assembled the four Sequel libraries together (termed Flye co-assembly), and then with the Sequel II library (termed Ultimate co-assembly) using metaFlye. Finally, we co-assembled the four Sequel libraries with the two Illumina metagenomes using metaSPAdes. Open reading frames (ORFs) of core metabolic genes were predicted from the assembled metagenomes using Prodigal[65] and annotated using Prokka[66] in KBase (https://kbase.us/)[67]. All assemblies were quality-checked using MetaQUAST[37] which precluded the inclusion of misassemblies from our analysis.

**Biosynthetic gene cluster analysis**. All contigs >5 kb in length were explored for biosynthetic gene clusters (BGCs) using antiSMASH v5.0 web server under strict settings[41]. Each predicted BGC was manually inspected for completeness to determine which were truncated on the contig edges as well as to investigate predicted chemical structures. Next, we consolidated and passed all putative BGCs through BiG-SCAPE v0.0.0r and CORASON in glocal mode to explore the phylogenomic relationships between the BGCs recovered from the 11 biocrust metagenomic datasets[14]. BiG-SCAPE consolidates both antiSMASH and the MiBIG 2.0 database to support initial antiSMASH predictions and so we included the entire MiBIG 2.0 database in our analysis to place our BGCs among verified clusters[68]. We checked for duplicate contigs containing BGCs across assemblies using BB-Dedup under default parameters and found no duplicates (sourceforge.net/projects/bbmap/).

To determine the genetic novelty of our BGCs we performed homology searches against the NCBI nt database (downloaded December 6th, 2019) using NCBI blast+ 2.9. We only retained top hits based on an E-value of $1 \times 10^{-10}$. BGCs were non-redundant (not sequenced previously and thus novel) if sequences matched ≤80% of the BGC query length and had an average of ≤75% sequence identity against the database. We corroborated the best BLAST taxonomic assignments with the Contig Annotation Tool under default settings (CAT, v5.0.4)

to guard against incorrect taxonomic identifications[69]. Chemical structure predictions were first created by antiSMASH v5.0.

**Metatranscriptomic mapping**. We made use of metatranscriptomes sequenced from biocrust material collected at the same sampling site in Moab, Utah that were publicly-available on JGI GOLD[47,48]. The experimental design tracked the transcriptional responses of biocrust communities over two complete diurnal cycles following an artificial wetting event in the laboratory with 12 h of light followed by 12 h of dark (Supplementary Data 1). The time points at which transcripts were collected include: 0 h (immediately before wet-up), 3 min, 15 min, 1 h, 9 h, 11.5 h, and 18 h after wet up, then 72 h after wet up (immediately prior to dry down), then 2 h and 3 days after dry down. The 11.5 h and 18 h samples also represent transcriptional activity at night-time while all other samples captured transcription during the day.

Transcripts were quality-controlled using Prinseq-lite v0.20.4 as described above for the Illumina data. The unassembled reads were then mapped to assembled metagenomic contigs using bbmap v38.73[50]. We then used SAMtools v1.9[70] for file conversion (sequence alignment maps to binary alignment maps) and sorting. The mapped sequences and their contigs were then visualized within Geneious[71] to assess mapping rates across the contig length.

**Statistics and reproducibility**. We used DESeq2 v1.28.0[72] under default parameters in the R statistical environment v3.6.3 to test which genes underwent differential expression by explicitly testing expression against the control sample (0 h). Here we tested two environmental treatments, (i) the diurnal cycling regime (i.e., day to night to day) and, independently, (ii) the influence of wetting and drying. Transcripts that had a maximum count of fewer than 20 reads in any sample were removed. The remaining transcript levels were normalized by the total counts for each sample and then multiplied by the average count across all samples. Duplicate samples at the 15-min time point and triplicate samples at the 1-h timepoint were averaged, and z-scores of normalized transcript abundance mapped to each biosynthetic gene to reveal which time points showed highest gene activity. Pearson two-sided pairwise correlations were calculated on normalized values using Pingouin (v 0.4.0) Python package with a one-step Bonferroni correction. Here, the average Z-score of mapped transcripts to contigs with BGCs were compared among phyla using the aforementioned filter of at least 20 mapped transcripts. The correlations consider the average Z-score pattern across all timepoints (Supplementary Data 8) after which Benjamini-Hochberg FDR correction is applied to the p-value statistic. In addition, z-scores were used with t-SNE (T-distributed Stochastic Neighbor Embedding) to visualize the biosynthetic gene transcription patterns in ordinance space[52]. The t-SNE implementation in sklearn (v 0.23.2) manifold module was used with the following parameters: "angle": 0.5, "early_exaggeration": 12.0, "init": "random", "learning_rate": 200.0, "method": "barnes_hut", 'metric": "euclidean", "min_grad_norm": 1e-07, "n_components": 2, "n_iter": 3000, "n_iter_without_progress": 300, "perplexity": 40, "random_state": None, "verbose": 1. We also mapped transcripts to the 16S rRNA gene data to estimate microbial transcription for each phylum. The metatranscriptomes were also assembled using metaSPAdes to explore whether entire BGCs could be recovered from this data.

**Publicly-available long-read metagenomes**. We downloaded four Sequel metagenomes (Pacific Biosciences) derived from Lake Biwa, Japan[40] as well as RS II and Illumina metagenomes from a biogas reactor[39] (Supplementary Data 1). All PacBio metagenomes were assembled with Canu v1.8[34] using the following parameters: genome size of 5 Mb (genomeSize = 5 m), corMinCoverage = 0, corOutCoverage = all, corMhapSensitivity = high, correctedErrorRate = 0.105, corMaxEvidenceCoverageLocal = 10, corMaxEvidenceCoverageGlobal = 10. metaSPAdes[36] was used for the short-read biogas metagenome and the co-assembly of both biogas metagenomes. Unfortunately, we were unable to assemble the biogas PacBio metagenome using either Canu or metaFlye[35], possibly owing to insufficient sequence coverage to generate contigs. We also co-assembled the four Lake Biwa Sequel metagenomes using Canu. The quality of each assembly was provided by MetaQUAST v4.6.3[37] and is available in Supplementary Data 2. We predicted BGCs from the assembled long-read metagenomes using antiSMASH v5.0[41]. The results of this analysis showed that, as for the biocrust metagenome assemblies, that co-assembling multiple metagenomes dramatically improves yield of BGCs. For example, the Lake Biwa samples produced 50 BGCs individually, but when co-assembled yielded 59 BGCs. This modest improvement was also observed in full-length BGCs (16 compared to 18).

**Reporting summary**. Further information on research design is available in the Nature Research Reporting Summary linked to this article.

## Data availability

Raw data of the long- and short-read biocrust metagenomes can be accessed on the IMG/M website (Submission ID 241874) or on the NCBI website (BioProject: PRJNA691698). The raw metatranscriptomic data are publicly-available through the JGI GOLD portal (sequence project IDs 1010318–1022409). Source data underlying Figs. 1b and 2a are presented in Supplementary Data 9, source data underlying Fig. 1c–e are presented in

Supplementary Data 3, source data underlying Fig. 2b are present in Supplementary Data 5, and source data underlying Fig. 3c are presented in Supplementary Data 10.

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

## Acknowledgements

This work was partially supported by funds provided by the Office of Science Early Career Research Program Office of Biological and Environmental Research, of the U.S. Department of Energy and by the U.S. Department of Energy Joint Genome Institute, a DOE Office of Science User Facility, supported by the Office of Science of the U.S. Department of Energy under Contract No. DE-AC02-05CH11231 to Lawrence Berkeley National Laboratory. We also wish to acknowledge Simon Roux, Emiley Eloe-Fadrosh and Eoin Brodie for their constructive feedback.

## Author contributions

M.W.V.G. conducted the metagenomic analyses, formulated ideas, and wrote the manuscript. A.R.O. predicted biosynthetic gene clusters, performed DESeq2 analysis and wrote the manuscript. B.P.B. conducted the statistical analyses among transcriptional profiles and produced figures. P.F.A. assisted with contig annotation and provided ideas. T.L.S. collected samples and extracted DNA. A.C. provided access to data, collaborated with Pacific Biosciences scientists to produce long-read metagenomes and offered valuable insights into the sequence data. R.R. assembled the large metagenomes and assisted with transcript mapping. G.H. developed technology of high-molecular weight DNA shearing and size selection optimization for library construction in collaboration with Pacific Biosciences scientists. M.K. developed Pacific Biosciences SMRTbell template preparation methods for HiFi 10 kb libraries from sample quality assessment to library preparation including development of multiplex strategy. L.S. and M.Y. optimized sequencing conditions for metagenomic sample libraries on the Pacific Biosciences RS II, Sequel and Sequel II sequencer platforms in collaboration with Pacific Biosciences. C.G.D. contributed sequencing platform development and operation management to enable the sequencing of environmental metagenomic samples on the Pacific Biosciences sequencing instruments in collaboration with Pacific Biosciences. Y.Y. developed long-read technology for environmental samples from design/coordination to execution in collaboration with Pacific Biosciences. T.P.M. offered logistical support and constructive feedback on the manuscript. F.G.P. provided access to *Microcoleus* genomes and assisted with meaningful interpretation of the cyanobacterial transcriptomics data. R.C.O. contributed overseeing the development of long-read technologies including 10 kb HiFi sequencing for environmental metagenomics in collaboration with Pacific Biosciences and JGI staff. A.V. wrote the paper and offered substantial advice on project planning. L.A.P. assisted in the interpretation of biological and genomic data. T.R.N. designed the study and wrote the manuscript.

## Competing interests

All authors read and approved the final manuscript. The authors declare no competing interests. Thulani P. Makhalanyane is an Editorial Board Member for *Communications Biology*, but was not involved in the editorial review of, nor the decision to publish this article.
