## [Transparent Peer Review File · Communications Biology]

Reviewers' comments:

Reviewer #1 (Remarks to the Author):

The manuscript describes the recovery of gene clusters for bioactive compounds that the authors have named "biosynthetic gene clusters" BGCs. They have explored an unusual environment a cyanobacterial-rich soil crust by a combination of 2nd and third-generation (long read) sequencing. There is great potential in the use of long reads in metagenomics and specifically in the recovery of large gene clusters that assemble poorly in Illumina generated MAGs, although not because they are produced by rare microbes but rather due to their presence in strain-specific (flexible) genomic islands. However, the approach taken here, although rewarding at the numbers of gene clusters recovered has several conceptual drawbacks. The fact that the total DNA was extracted might have affected the quality of the assembly given the potential presence in the sample of eukaryotic (e.g., fungal) DNA. The figures of assembly size are rather poor compared to other studies carried out with or without long reads. The main result of this work is the large volume of BGCs recovered and their diversity. However, this topic has been barely touched. Instead, the authors expand the work with a rather unwarranted metatranscriptomic analysis. They justify this study by the frequent lack of expression of BGCs in the lab, which is certainly true, but their study has been carried out mimicking natural conditions with wetting of the soil and exposure to dial cycles.

Ln 53. It is not true that the dominant members of the community are recovered as MAGs, it is often the reverse due to the high diversity of clonal lineages present that complicate assembly. On the other hand, the authors do not mention the difficulty in the assembly of the flexible (sometimes called accessory) genome that contains often the BGCs.

Ln 76. Again, assembly is not related to the "rareness" of microbes. Some rare microbes assemble much better than abundant ones. Subsampling is routine for improving assembly decreasing coverage.

Ln 83. "intact biocrust" will be composed also of eukaryotic cells like fungi or even microscopic metazoa such as rotifers or worms (and their microbiomes), this fact should be considered throughout (18S rRNA detection would be helpful, Ln 90).

Ln 109 >50 Kb doesn't sound like "ultra-long" but of course, it is subjective.

Ln 143 the biocrust is a "rich source" compared to what, the authors do not describe similar approaches applied to any other habitat.

Ln 163 Again, 18S rRNAs should be analyzed together with 16S

Ln253 time before DNA extraction should also be provided.

Ln 257 Please describe measures to increase DNA fragment size.

Ln 156 (whole section). I have found this section puzzling and hard to follow. From the methods it seems that transcripts were assembled, and the individual reads compared to the assemblies of BGCs. The usual strategy would be to compare the transcriptomic reads to the assembled metagenomes. It might be a misunderstanding but then the results are described in a superficial and often confusing way. First, there is the claim that 6% of the BGSs are constitutively transcribed but there is no discussion of which ones or whether the data for all the genes in the cluster were consistent. Second, the list of BGCs as shown in Figure 1 is extremely heterogeneous and it seems dangerous to put them all together for the transcription analysis when it is obvious that the biological role of each group could be dramatically different. The assumption that they are all involved in antagonistic interactions is

rather naïve. The experiment seems also to be missing some controls, for example, the difference between day and night in the dry soil crust.

Reviewer #2 (Remarks to the Author):

The paper by Goethem et al. reports close to 3,000 BGCs from biocrust, 695 of which are novel and full length BGCs. They use short and long-read metagenomic sequencing approaches and further claim that by using long-read metagenomic sequencing one can recover BGCs that would otherwise be missed, especially from rare biospheres. I believe the paper is novel just based on the number of novel BGCs reported, use of long-read metagenomes, and the wetting experiment that provided further insights into the expression profile of these BGCs in their natural environments. However, the paper was rather confusing to read in multiple places, especially on why some of the analyses were done and if those analyses contributed or even enhanced the main findings. Additionally, I think authors are in the position of commenting on how well this approach will fare when using it on complex microbial communities. Here in the Major Comments section I have tried to describe sources of my confusions.

Major comments to address:

In the abstract authors mention they recovered 695 novel, full-length BGCs, but in the main paper, I cannot find the source of that number. As a matter of fact 695 only appears in the abstract, but nowhere else. I think this is rather important to clarify. Authors do mention that they recovered a total of 712 full-length BGC clusters and I am guessing they were able to find a hit in NCBI-nr for 17 of them and that's how they came up with 695. However, this information is not clearly mentioned anywhere in the manuscript.

Another confusion comes from multiple iterations of assemblies, co-assemblies, and hybrid assemblies that were performed, but not clearly describing, which of the approach was used for downstream assessments and if combined, how they were combined. For example, how did the authors find 712 BGCs? Were they combined from all the iterations of assemblies? If so, were the BGCs checked if some of them are duplicates? A main table that summarizes samples, runs, and assemblies could be helpful here.

In the paper, authors claimed that 94% of the BGCs that they recovered have never been sequenced before. To prove this claim they search BGCs reported in the paper against the NCBI nt database. Does this consider MAGs? Or is MAGs represented in the nt database? I think it might make sense to modify the claim accordingly to reflect what's represented in the nt database or expand the search to at least available MAGs.

Minor comments to address:

- Please mention how many biocrust samples were collected and sequenced clearly in the main paper. Maybe in line 81 as it is not clear if all the sequencing were done from one or multiple biocrust samples.

Line 82, Second sentence of this paragraph seems grammatically incorrect.

In Line 92, the authors claim that the cyanobacteria are the dominant biocrust community members but figure 1b shows it may only be dominant in 4/9 samples.

Can the authors explain how they predicted 712 BGCs as full-length clusters? Were they all manually checked like the hybrid trasnAT-PKS-NRPS BGC? Also, can the authors explain what signatures or features were checked during these manual verifications?

Line 133: I think just BGC is sufficient here.

Line 132: 2988 out of how many total contigs.

Line 158, should it be RNA microarray?

Line 267, where are the results of these analyses

Line 283, how were the taxonomies of gene clusters resolved if the sequences hit multiple genomes

during BLAST.

Reviewer #3 (Remarks to the Author):

For this study the authors leveraged long- and short-read sequencing to investigate biosynthetic gene clusters (BGCs) in microbiomes found in biological soil crusts. Different methods for sequencing, library preparation and assembly were used to compile a large collection of BGCs. A majority of these gene clusters did not produce good matches during an NCBI nt database search and can thus be classified as newly sequenced. Expanding the amount of known BGCs is likely to be of use for future research.

To gain insight into the production of secondary metabolites, metatranscriptome data from a time-series experiment carried out in the context of a previous study were utilized. Reads were mapped to the assembled BGCs to investigate how the activity of taxa as well as the transcription of BGCs changes in response to varying environmental conditions. Statistical analysis of this data points to a higher-than-expected transcription of BGCs over the time series with peak transcriptional activity at night. Results show that Cyanobacteria were dominant in terms of nightly BGC transcription and the authors postulate a connection to suspended photosynthetic activity in Cyanobacteria at night. Investigating BGC expression at phylum-level reveals distinct expression profiles for the different phyla, possibly related to their different niche competition strategies. The study furthers understanding of the role of biosynthetic gene clusters and secondary metabolism products in soil microbial communities.

In its current form the manuscript does not provide sufficiently detailed information to reproduce the work of the authors. For this reason the soundness of the statistical methods cannot be fully assessed. The specific comments below address the aspects of the study which are in need of further explanation.

1. (Line 82-83) Spelling/grammar. Double `extracted and prepared`. It should be: "We then extracted and prepared high-molecular-weight DNA from intact [...]"
2. (Line 103 & 106) What is the reason for using only five (of eight) metagenome datasets? What was the selection criteria for choosing the four long read datasets that were hybrid-assembled with the Illumina data?
3. (Line 141) The used NCBI database is old. How would the results change if you used a new one?
4. (Line 171) Please add parameters of bbmap use.
5. (Line 174) Is there a motivation for the cutoff at five mapped reads per gene? The number 5 seems pretty arbitrary.
6. (Line 175, Line 189, Fig. 2b) Labeling of the y-axis in figure 2b suggests that it indicates which percentage of all RNA reads mapped to genes of the respective category. However, the text in the aforementioned lines specifies the percentage of genes which show expression that clears a certain threshold. Please clarify what is shown in the figure and adjust the labeling if necessary.
7. (Line 182) Please clarify: Is this observation of other genes based on mapping against BCG-containing contigs only?
8. (Line 187) Spelling/grammar. "[...] focused on 46 core metabolic [...]"
9. (Line 201) Is this an adjusted p value (multiple test correction)?
10. (Line 207) Please explain further the correlation that was measured here. What data served as input?
11. (Line 220) Which test produced this p value?
12. (Line 247 to 257) The Materials and Methods section does not clearly explain whether there are biological differences in the samples that were used in the various sequencing experiments. Please

elucidate: Are all sequencing experiments based on the same biological sample? If yes, please address the different results regarding taxonomic composition (shown in Fig. 1b). If not, please indicate which sequencing library is associated with which biological sample.

13. (Line 332) What purpose does the metatranscriptome assembly serve? Supplement S5 suggests that a search for BGCs was carried out in the MT-assembly, but this is not described, only mentioned briefly in line 54 of the supplement text. Table S2 suggests the existence of several different metatranscriptome assemblies for different time points, but again, this is not mentioned in the text.

14. (Line 333) Suggest replacing "transcripts" with "reads" for clarity.

15. (Line 333) It is my understanding that RNA reads were only mapped to metagenome contigs containing biosynthetic gene clusters. Using a complete assembly for the mapping step would keep reads from erroneously being mapped to BCGs if the mapper can find a better alignment on a non-BCG region. Please elaborate on the reason for choosing only the BCG-contigs as targets. Please provide details regarding the parameters that bbmap was used with.

16. (Line 335) To facilitate reproducibility, please elaborate on the use of DESeq2 (design formula and any non-default parameters used like type of differential expression test or type of multiple comparison adjustment)

17. (Line 339) Please provide details regarding this step. What is meant by transcripts mapping to 16S Data?

18. (Line 389) BioProject accession is invalid, correct accession is PRJNA691698.

19. Please provide a citation regarding the metatranscriptome sequencing. The paper by Rajeev et al. (Reference 40) provides details about the biological samples, but does not mention RNA-sequencing.

20. (Figure 2c) Predicted structures are not shown on the right as the description of the figure would indicate. The description lacks an explanation of what exactly is shown between the heatmaps. Most importantly, it is unclear what the individual rows in the heatmap represent. Assuming each row represents one of the 13 time points, what is the base transcription level used to calculate fold changes? Or is the heatmap displaying transformed count data?

21. Tables S4 and S6 are delivered in a PDF format which makes them very difficult to read. Please provide these as xlsx files like the other tables in the supplement.

22. (Figure S7) It is unclear what is shown here (e.g. fold changes, p-values). Please provide additional details in the description of the figure and add a label to the color chart.

23. (Figure S8) Lacks labeling of the color charts.

24. sometimes, the language seems a bit ultra-exaggerating ("ultra-long contigs", "ultra-large assemblies", "ultra-long gene clusters", etc.)

Reviewers' comments:

Reviewer #1 (Remarks to the Author):

We thank the reviewer for their useful comments. Based on these comments we have added additional discussion of the limitations of our work, such as the lack of information on eukaryotic diversity, and clarified our metatranscriptomic analyses to be more succinct. Moreover, our oversight that most secondary metabolites would cause negative interactions has been amended, and this comment led us to explore potentially synergistic effects as well, which we appreciate as this information was useful in part of the improved discussion.

The manuscript describes the recovery of gene clusters for bioactive compounds that the authors have named "biosynthetic gene clusters" BGCs. They have explored an unusual environment a cyanobacterial-rich soil crust by a combination of 2nd and third-generation (long read) sequencing. There is great potential in the use of long reads in metagenomics and specifically in the recovery of large gene clusters that assemble poorly in Illumina generated MAGs, although not because they are produced by rare microbes but rather due to their presence in strain-specific (flexible) genomic islands. However, the approach taken here, although rewarding at the numbers of gene clusters recovered has several conceptual drawbacks.

Thank you for recognizing the potential of this approach. We have made extensive changes to the manuscript to address your comments as detailed below.

The fact that the total DNA was extracted might have affected the quality of the assembly given the potential presence in the sample of eukaryotic (e.g., fungal) DNA.

You are correct. Since more developed biocrust are known to harbor fungi, among other less abundant organisms, it is likely that a low diversity of fungal species are present in early-stage biocrust too. Naturally, their larger genomes would complicate downstream assembly as their sequence signals would dwarf bacterial sequences due to smaller genome sizes. However, since bacteria are typically smaller, there are often more copies of their genomes in microbiome samples as they occupy more soil than fungi. In addressing this, and other similar comments by this reviewer, we have assessed the diversity of 18S rRNA and ITS gene sequences in these biocrust metagenomes. Yet we recovered very few eukaryotic genes, in the order of 10 – 100 sequences, from these metagenomes, which is largely consistent with research on early-stage biocrust that found low levels of eukaryotic signal in their samples at these sites in Moab, Utah (see Van Goethem et al., 2018 - Characteristics of wetting-induced bacteriophage blooms in

biological soil crust). Taking advantage of the unusually long assemblies produced by this approach, we recovered nearly 3,000 BGCs for analysis, including 712 novel, full-length BGCs.

We have, however, noted this as a limitation of our work.

Ln 100: “We found very few sequences for eukaryotes after seeking 18S rRNA genes, range 100 – 1000 sequences per metagenome, and acknowledge that more attention should be given to the diverse members of kingdom in future research as fungi, rotifers, algae and mosses can constitute important components of the biocrust microbiome (Büdel, Dulić et al. 2016, Bowker, Reed et al. 2018). While this work was focused on BGCs for bacteria, we saw a small number of eukaryotic reads (>1% of all sequences, although no BGCs) and this would be an interesting investigation for a future analysis of this dataset, given their importance in late-stage biocrust (Bowker, Reed et al. 2018, Giraldo-Silva, Nelson et al. 2019).”

The figures of assembly size are rather poor compared to other studies carried out with or without long reads.

We agree with this criticism of long-read sequencing and assembly. Put our work in context of other metagenomes. With the following sentences:

Ln 131: “To put our assemblies in context of existing sequence technology, our long-read co-assembly is ~90% of the very large short-read Iowa prairie metagenomes at 1.5 billion bp (Howe, Jansson et al. 2014), but 1,473% and 1,688% of the biogas and Lake Biwa long-read co-assemblies, respectively (Frank, Pan et al. 2016, Hiraoka, Okazaki et al. 2019). Despite their smaller size compared to short-read metagenome assemblies, we get a higher percentage of long contigs from long-read metagenomes which is paramount for BGC detection.”

The advantage of long-read sequencing is undoubtedly read length, yet this comes at the cost of sequencing effort (i.e. size). Downstream, the long-read assemblies are generally smaller than those generated using short-read sequencing since the assemblers don't prioritize the assembly of many short contigs, and instead provide fewer, longer contigs. Thus, the metric of assembly size is poor for long-read metagenome assembly, but this is not due to a dearth of sequence information and actually reflects the prioritization of longer contigs.

In our dataset, the Sequel-Sequel II co-assembly has the most sequence in contigs >50,000 bp (499,106,635 bp) from only 1,352,413,409 bp of total sequence information. Thus, more than 1/3 of the sequence is present in long contigs (36.9%). For Illumina assemblies there is only 1,621,809 bp present in long contigs (50,000 bp) out of 1,941,581,580 bp total assembly (0.08%).

Moreover, the assembly sizes are larger than both Lake Biwa and the Biogas long-read metagenome assemblies that we also analyzed (Table S2).

The main result of this work is the large volume of BGCs recovered and their diversity. However, this topic has been barely touched. Instead, the authors expand the work with a rather unwarranted metatranscriptomic analysis. They justify this study by the frequent lack of expression of BGCs in the lab, which is certainly true, but their study has been carried out mimicking natural conditions with wetting of the soil and exposure to diel cycles.

This is a valid criticism and reflects our own particular scientific goals. Specifically, our goal here was to use this long-read technology to survey the ecology of the BGCs though complementing BGC discovery with information about their transcription. We feel that other groups with specific expertise in natural product discovery are better suited to perform an in-depth exploration of the diversity of BGCs recovered using such an approach.

We have now noted this in the revised text.

Ln 164: “We recognize that there is significant opportunity to do more in-depth analyses of the diversity of BGCs recovered and feel that this presents a promising future research direction.”

Ln 53. It is not true that the dominant members of the community are recovered as MAGs, it is often the reverse due to the high diversity of clonal lineages present that complicate assembly. On the other hand, the authors do not mention the difficulty in the assembly of the flexible (sometimes called accessory) genome that contains often the BGCs.

This is a good point. This is an overstatement and therefore we have revised the text to:

Ln 52: “Although these approaches can give access to both dominant and rare members of the community, many contigs will not be binned into a genome (Lynch and Neufeld 2015). Moreover, long-read sequencing circumvents the requirement of generating MAGs in some cases, as large genome segments are captured directly through sequencing and assembly, which could favor low-abundance species.”

Ln 76. Again, assembly is not related to the “rareness” of microbes. Some rare microbes assemble much better than abundant ones. Subsampling is routinary for improving assembly decreasing coverage.

Thank you for this comment. To avoid confusion here we simply removed “both rare and dominant microbes” and replaced with “uncultivated microbes” on Ln 79.

Ln 83. "intact biocrust" will be composed also of eukaryotic cells like fungi or even microscopic metazoa such as rotifers or worms (and their microbiomes), this fact should be considered throughout (18S rRNA detection would be helpful, Ln 90).

Yes, this remains a good point. As mentioned earlier, we have included 18S rRNA gene analysis in the latest version of the manuscript, yet found very low levels of signal for Eukaryotes which is consistent with the early-stage biocrust that we have analyzed.

Ln 100: "We found very few sequences for eukaryotes after seeking 18S rRNA genes, range 100 – 1000 sequences per metagenome, and acknowledge that more attention should be given to the diverse members of kingdom in future research as fungi, rotifers, algae and mosses can constitute important components of the biocrust microbiome (Büdel, Dulić et al. 2016, Bowker, Reed et al. 2018)."

Ln 109 >50 Kb doesn't sound like "ultra-long" but of course, it is subjective.

Reviewer 3 had a similar concern. We have changed it to "long" contigs throughout the paper including on Lns 75, 83, 121 and 136.

Ln 143 the biocrust is a "rich source" compared to what, the authors do not describe similar approaches applied to any other habitat.

Thank you for spotting this omission. We have added a comparison to other systems to justify this claim by linking the reader to our supplementary results on Lake Biwa and a biogas reactor in which far fewer BGCs were recovered than in biocrust.

Ln 161: "This reaffirms that biocrust are a rich source of BGCs relative to either aquatic systems such as Lake Biwa or biogas reactors which are simpler systems compared to soil (see supplemental material). Moreover, these results and underscores the potential for long-read metagenomic sequencing in novel BGCs discovery."

Ln 163 Again, 18S rRNAs should be analyzed together with 16S

Yes, we have done so and found very low signals of 18S rRNA genes.

Ln253 time before DNA extraction should also be provided.

Good catch. This has been added.

Ln 299: "Samples were maintained in petri dishes in a dark desiccator in the laboratory until required for DNA isolation (approximately 2 months)."

The dried biocrusts were stored in a cool and dry location for 2 months prior to DNA extraction.

Ln 300: "Previous studies have shown that biocrusts are viable for long-periods of times when stored under these conditions (Kidron and Tal 2012, Fernandes, Machado de Lima et al. 2018, Karaoz, Couradeau et al. 2018, Giraldo-Silva, Nelson et al. 2019)."

Ln 257 Please describe measures to increase DNA fragment size.

Thank you for pointing this out. Oversight that we didn't mention the following, now in text:

Ln 311: "This biocrust sample was well-suited for long-read sequencing as it contained high-quality DNA without the need for additional steps to increase DNA fragment size."

Ln 156 (whole section). I have found this section puzzling and hard to follow. From the methods it seems that transcripts were assembled, and the individual reads compared to the assemblies of BGCs. The usual strategy would be to compare the transcriptomic reads to the assembled metagenomes. It might be a misunderstanding but then the results are described in a superficial and often confusing way.

We apologize for the confusion in this section, and we have revised and reorganized the text including additional detail. You are entirely correct, the unassembled transcripts were mapped to the assembled metagenomes. We later assembled the transcripts to search for BGCs encoded by this dataset. We hope this is clearer. The methods section better clarifies our approach and now reads as follows:

Ln 385: "Transcripts were quality-controlled using Prinseq-lite v0.20.4 as described above for the Illumina data. The unassembled reads were then mapped to assembled metagenomic contigs using bbmap v38.73 (Bushnell 2014). We then used SAMtools v1.9 (Li, Handsaker et al. 2009) for file conversion (sequence alignment maps to binary alignment maps) and sorting. The mapped sequences and their contigs were then visualized within Geneious (Kearse, Moir et al. 2012) to assess mapping rates across the contig length. Next, we used DESeq2 v1.28.0 (Love, Huber et al. 2014) under default parameters in the R statistical environment v3.6.3 to test which genes underwent differential expression by explicitly testing expression against the control sample (0 hours). Here we tested two environmental treatments, (i) the diurnal cycling regime (i.e., day to night to day) and, independently, (ii) the influence of wetting and drying. Transcripts that had a

maximum count of fewer than 20 reads in any sample were removed. The remaining transcript levels were normalized by the total counts for each sample and then multiplied by the average count across all samples. Duplicate samples at the 15-minute time point and triplicate samples at the 1-hour timepoint were averaged, and z-scores of normalized transcript abundance mapped to each biosynthetic gene to reveal which time points showed highest gene activity. Pearson two-sided pairwise correlations were calculated on normalized values using Pingouin (v 0.4.0) Python package with a one-step Bonferroni correction. Here, the average Z-score of mapped transcripts to contigs with BGCs were compared among phyla using the aforementioned filter of at least 20 mapped transcripts. The correlations consider the average Z-score pattern across all timepoints (Table S8) after which Benjamini-Hochberg FDR correction is applied to the p-value statistic. In addition, z-scores were used with t-SNE (T-distributed Stochastic Neighbor Embedding) to visualize the biosynthetic gene transcription patterns in ordinance space (Van Der Maaten 2014). The t-SNE implementation in sklearn (v 0.23.2) manifold module was used with the following parameters: 'angle': 0.5, 'early_exaggeration': 12.0, 'init': 'random', 'learning_rate': 200.0, 'method': 'barnes_hut', 'metric': 'euclidean', 'min_grad_norm': 1e-07, 'n_components': 2, 'n_iter': 3000, 'n_iter_without_progress': 300, 'perplexity': 40, 'random_state': None, 'verbose': 1. We also mapped transcripts to the 16S rRNA gene data to estimate microbial transcription for each phylum. The metatranscriptomes were also assembled using metaSPAdes to explore whether entire BGCs could be recovered from this data.”

First, there is the claim that 6% of the BGSs are constitutively transcribed but there is no discussion of which ones or whether the data for all the genes in the cluster were consistent.

We have added more discussion to describe this observation in sufficient depth. The 6% value is based on all contigs containing BGCs. We had added this description to the text.

Ln 202: “Most of the constitutively transcribed genes within BGCs were AMP-binding domains, condensation domains, PCP domains and phytoene synthases, all of which have central roles in the biosynthesis of specialized metabolites. *Cyanobacteria*, and to a lesser extent *Actinobacteria*,

encoded most of these genes while our results further indicate that NRPS and PKS gene clusters were the most prominent class of secondary metabolites that were constitutively transcribed. In some cases we observed that multiple consecutive genes within the same BGC were constitutively transcribed. On average we found that at least two genes on the contig were constitutively transcribed while single genes within a BGC experiencing constitutive transcription were rare.”

Second, the list of BGCs as shown in Figure 1 is extremely heterogeneous and it seems dangerous to put them all together for the transcription analysis when it is obvious that the biological role of each group could be dramatically different.

This is true, and there is substantial variability in the types and quantities of BGCs recovered here. We have clarified our goal in this step of the analysis. The aim of grouping all, unrelated BGCs was to explore whether trends in their transcription would emerge, while not necessarily exploring their biological roles simultaneously. For example, perhaps terpenes might be transcribed at a specific time point, or that *Cyanobacteria* only transcribed their siderophores-encoding BGCs at night. Without doing this we would have limited our scope of observation in our opinion.

We have clarified this in the manuscript.

Ln 177: “Here we opted to analyze all BGCs simultaneously in an effort to unearth variations between their transcriptional profiles that could emerge following wetting, for example.”

The assumption that they are all involved in antagonistic interactions is rather naïve.

We agree with this assessment, we have made it clear that BGC encode for diverse secondary metabolites involved in a variety of cellular processes, few of which are produced exclusively for competitive interactions among microbes.

Ln 269: “Thus, at night *Microcoleus* and other cyanobacteria may produce antibiotics, such as bacteriocins that we detected here, to antagonize heterotrophs competing for dissolved organic compounds (Kupriyanova, Sinetova et al. 2011), although we note that many other secondary metabolic functions may be performed at night including redox balancing or siderophore and pigment production.”

The experiment seems also to be missing some controls, for example, the difference between day and night in the dry soil crust.

We wholeheartedly agree. The insights we might have obtained under those conditions would have certainly strengthened our analyses. The metatranscriptomic data are from previous studies (Rajeev *et al.*, 2013, Nunes da Rocha *et al.*, 2015) which we had no control over, unfortunately.

Reviewer #2 (Remarks to the Author):

We thank this reviewer for their effort in reviewing our manuscript and the many helpful comments. We have revised the manuscript to address the points you raised and feel the clarity of the manuscript has been greatly improved. Particularly the introduction of the new table (Table 1) describing the assemblies and the BGCs recovered from each approach has been useful in clarifying the manuscript.

The paper by Van Goethem et al. reports close to 3,000 BGCs from biocrust, 695 of which are novel and full length BGCs. They use short and long-read metagenomic sequencing approaches and further claim that by using long-read metagenomic sequencing one can recover BGCs that would otherwise be missed, especially from rare biospheres. I believe the paper is novel just based on the number of novel BGCs reported, use of long-read metagenomes, and the wetting experiment that provided further insights into the expression profile of these BGCs in their natural environments. However, the paper was rather confusing to read in multiple places, especially on why some of the analyses were done and if those analyses contributed or even enhanced the main findings. Additionally, I think authors are in the position of commenting on how well this approach will fare when using it on complex microbial communities. Here in the Major Comments section I have tried to describe sources of my confusions.

We thank the reviewer for their enthusiasm for our manuscript.

Major comments to address:

In the abstract authors mention they recovered 695 novel, full-length BGCs, but in the main paper, I cannot find the source of that number. As a matter of fact 695 only appears in the abstract, but nowhere else. I think this is rather important to clarify. Authors do mention that they recovered a total of 712 full-length BGC clusters and I am guessing they were able to find a hit in NCBI-nr for 17 of them and that's how they came up with 695. However, this information is not clearly mentioned anywhere in the manuscript.

We apologize for this confusion, and thank the reviewer for catching this critical error. This value was determined by manually checking each BGC for truncation on either (or both) of the contig edges, but this value seems to be a carry-over from a previous version of the manuscript. You are correct and we have updated the abstract to 712 full-length clusters (Ln 24) which is consistent with the value in the rest of the text and in the supplementary table.

Ln 24: “Taking advantage of the unusually long assemblies produced by this approach, we recovered nearly 3,000 BGCs for analysis, including 712 full-length BGCs.”

Another confusion comes from multiple iterations of assemblies, co-assemblies, and hybrid assemblies that were performed, but not clearly describing, which of the approach was used for downstream assessments and if combined, how they were combined. For example, how did the authors find 712 BGCs? Were they combined from all the iterations of assemblies? If so, were the BGCs checked if some of them are duplicates? A main table that summarizes samples, runs, and assemblies could be helpful here.

Thank you for pointing this out. We have added a table to summarize our workflow to avoid this confusion in future. The total 2,989 BGCs are the sum total from all assemblies, co-assemblies and hybrid assemblies. Of these, 712 were full-length BGCs. Finally, we have run BB-Dedup (default parameters) to check whether duplicate contigs formed part of this analysis but found that none were duplicates. This has been clarified through the following revisions to the manuscript:

Ln 365 we have added: “We checked for duplicate contigs containing BGCs across assemblies using BB-Dedup under default parameters and found no duplicates (sourceforge.net/projects/bbmap/).”

Ln 129 we added: “In total, that is, including all assemblies, we predict that 712 BGCs are full-length clusters.”

Ln 149 we have also added: “This approach recovered 2,988 BGCs predicted to produce secondary metabolites from uncultivated biocrust microbes across all metagenome assemblies.”

Ln 161 we have added Table 1 to capture the aforementioned information.

A	B	C	D	E	F
Co-assembly	Number of BGCs	Full-length BGCs	No. sequenced Previously	Sequencing Platforms	Metagenomes used for co-assembly/assembly
2	300	67	28	PacBio Sequel (n=4)	m54017_180413_173154, m54017_180414_134655, m54017_180414_220614, m54017_180417_205359
3	459	31	42	PacBio Sequel (n=4) & Illumina HiSeq 2500 (n=2)	m54017_180413_173154, m54017_180414_134655, m54017_180414_220614, m54017_180417_205359, 11774.4.218925, 12041.5.235284
4	1191	419	21	PacBio Sequel (n=4) & PacBio Sequel II (n=1)	m54017_180413_173154, m54017_180414_134655, m54017_180414_220614, m54017_180417_205359, pbio-2210.20021
5	16	0	3	PacBio RS II	pbio-1768.15750
5	5	0	1	PacBio RS II	pbio-1772.15782
7	0	0	0	PacBio RS II	pbio-1768.15751
3	24	3	8	PacBio Sequel	m54017_180413_173154
9	30	2	7	PacBio Sequel	m54017_180414_134655
0	0	0	0	PacBio Sequel	m54017_180414_220614
1	44	7	5	PacBio Sequel	m54017_180417_205359
2	548	174	9	PacBio Sequel II	pbio-2210.20021
3	139	3	19	Illumina HiSeq 2500	11774.4.218925
4	232	6	32	Illumina HiSeq 2500	12041.5.235284
5					

In the paper, authors claimed that 94% of the BGCs that they recovered have never been sequenced before. To prove this claim they search BGCs reported in the paper against the NCBI nt database. Does this consider MAGs? Or is MAGs represented in the nt database? I think it might make sense to modify the claim accordingly to reflect what's represented in the nt database or expand the search to at least available MAGs.

Good question. We did not explicitly search MAGs. That said, any MAGs submitted to the NCBI would appear in our queries in the nt database which collates all nucleic acid submissions. As for biocrust MAGs specifically, our earlier work (*Characteristics of wetting-induced bacteriophage blooms in biological soil crust*) is to date the only work on biocrust that has produced MAGs. The MAGs identified therein do not contain many BGCs (they are assembled from Illumina sequences and thus have shorter contigs), while the pioneer biocrust species, such as *Microcoleus vaginatus*, has multiple whole genome sequences available online, these only bare some sequence similarity to BGCs recovered by our work presented here. The same approach used by (Nayfach, Camargo et al. 2021) recovered 83% novel BGCs in an analysis of >50,000 MAGs, although this considered all biomes, most of which are more extensively sequenced than biocrust.

Minor comments to address:

Please mention how many biocrust samples were collected and sequenced clearly in the main paper. Maybe in Ln 81 as it is not clear if all the sequencing were done from one or multiple biocrust samples.

Good point. We have revised the text to make it clear that all the long-read metagenomic sequencing was done on a single biocrust community.

Ln 85 now reads: “high-molecular-weight DNA from an intact biocrust sample to be sequenced on both long- and short-read metagenomic sequencing platforms.”

Ln 82, Second sentence of this paragraph seems grammatically incorrect.

Yes this is wrong, we have altered this sentence to:

Ln 85: “We then extracted and prepared high-molecular-weight DNA from an intact biocrust sample to be sequenced on both long- and short-read metagenomic sequencing platforms (Fig. S1).”

In Ln 92, the authors claim that the cyanobacteria are the dominant biocrust community members but figure 1b shows it may only be dominant in 4/9 samples.

Great point, thank you. This highlights the variability among sequencing runs of the same sample. Indeed it is well known that cyanobacteria dominate these early-successional stage biocrust, yet the recovery of their full-length 16S rRNA genes from the assembled metagenomes is generally inconsistent and seems to have skewed estimates of taxonomy. For clarity we have changed this sentence to the following:

Ln 95: “*Cyanobacteria*, and particularly *Microcoleus vaginatus*, were dominant biocrust community members in most samples”

*Can the authors explain how they predicted 712 BGCs as full-length clusters? Were they all manually checked like the hybrid *trsnAT-PKS-NRPS* BGC? Also, can the authors explain what signatures or features were checked during these manual verifications?*

We determined whether BGCs were full-length by manually inspecting each contig containing a BGC. Firstly, we checked whether BGC was truncated on either contig edge, i.e. intersected the sequence boundary. Next, we manually checked the validity of known BGC predictions to identify structures that were incomplete. To clarify this point we have revised the text to state the following:

Ln 359: “Each predicted BGC was manually inspected for completeness to determine which were truncated on the contig edges as well as to investigate predicted chemical structures.”

Ln 133: I think just BGC is sufficient here.

We agree and have changed to the acronym here.

Ln 149 now reads: “This approach recovered 2,988 BGCs predicted to produce secondary metabolites from uncultivated biocrust microbes across all metagenome assemblies.”

Ln 132: 2988 out of how many total contigs.

This is out of 141,762 total contigs, we have added this information:

Ln 148: “high proportion of contigs longer than 5 kb ($n = 141,762$ total contigs; Fig. 1c)”

Ln 158, should it be RNA microarray?

Ln 183: Yes, thank you for catching this error. Changed to “RNA microarray”

Ln 267, where are the results of these analyses

These are in the supplementary material (we have added a link to this information to the main text), specifically to indicate that biocrust are rich sources of BGCs. These changes were made as follows:

Ln 318: “To test how well-suited long-read metagenomes are for BGC recovery, we also made use of five publicly available PacBio SMRT metagenomes. The results of these analyses are including in the Supplementary Results. The two datasets analyzed include a biogas reactor library sequenced on the PacBio RS II System with a 2 kb insert length, and four metagenomes

obtained from Lake Biwa, Japan that were sequenced on a PacBio Sequel System with a 4 kb insertion length (Supplementary Material).”

Ln 283, how were the taxonomies of gene clusters resolved if the sequences hit multiple genomes during BLAST.

More often than not the top hits match the same species, or at worst the same genus. Regardless, we selected the top hit (most sequence similarity across most of the sequence) and corroborated these results with the contig annotation tool (CAT). We have added this to the main text:

Ln 372: “We corroborated the best BLAST taxonomic assignments with the Contig Annotation Tool under default settings (CAT, v5.0.4) to guard against incorrect taxonomic identifications.”

Reviewer #3 (Remarks to the Author):

We thank the reviewer for making the effort to provide us with many helpful comments. We have worked to carefully address the many issues you have identified and feel that the manuscript is greatly improved. Specifically, the confusions you had about the types of assemblies we performed have been amended which certainly clarifies our points. Finally, the suggestions by this reviewer for more detailed methodologies improve the reproducibility of our workflow and should aid readers in replicating our approach on their own datasets.

For this study the authors leveraged long- and short-read sequencing to investigate biosynthetic gene clusters (BGCs) in microbiomes found in biological soil crusts. Different methods for sequencing, library preparation and assembly were used to compile a large collection of BGCs. A majority of these gene clusters did not produce good matches during an NCBI nt database search and can thus be classified as newly sequenced. Expanding the amount of known BGCs is likely to be of use for future research. To gain insight into the production of secondary metabolites, metatranscriptome data from a time-series experiment carried out in the context of a previous study were utilized. Reads were mapped to the assembled BGCs to investigate how the activity of taxa as well as the transcription of BGCs changes in response to varying environmental conditions. Statistical analysis of this data points to a higher-than-expected transcription of BGCs over the time series with peak transcriptional activity at night. Results show that Cyanobacteria were dominant in terms of nightly BGC transcription and the authors postulate a connection to suspended photosynthetic activity in Cyanobacteria at night. Investigating BGC expression at phylum-level reveals distinct expression profiles for the different phyla, possibly related to their different niche competition strategies. The study furthers understanding of the role of biosynthetic gene clusters and secondary metabolism products in soil microbial communities.

Thank you for the enthusiasm for our work

In its current form the manuscript does not provide sufficiently detailed information to reproduce the work of the authors. For this reason the soundness of the statistical methods cannot be fully assessed. The specific comments below address the aspects of the study which are in need of further explanation.

1. (Ln 82-83) Spelling/grammar. Double `extracted and prepared`. It should be:

Thank you for spotting this error. This has been corrected.

Ln 85: “We then extracted and prepared high-molecular-weight DNA from an intact biocrust sample to be sequenced on both long- and short-read metagenomic sequencing platforms”

2. (Ln 103 & 106) *What is the reason for using only five (of eight) metagenome datasets? What was the selection criteria for choosing the four long read datasets that were hybrid-assembled with the Illumina data?*

This selection was partially due to both time and computational resources (the Sequel II assembly was challenging to assemble alone, even without other datasets), but we also selected these metagenomes for co-assembly as the data in the supplemental material (Lake Biwa and biogas reactor datasets) had used PacBio RSII and Sequel data could be analyzed in tandem with Illumina metagenomes sequences. We aimed to compare these datasets by way of complimentary co-assembly approaches. Thus, we omitted co-assembling the Sequel II data with any other metagenomes.

We have revised this text to make this clear to the reader:

Ln 114: “Firstly, we co-assembled the five largest long-read metagenomes (Sequel and Sequel II metagenomes) which yielded 1.4 Gb of assembled sequence (Table S2) with the longest contig exceeding 1.3 Mb in length ($N50 = 36$ kb). We omitted the RS II metagenomes from co-assembly since their sizes are small compared to the Sequel counterparts.” We also generated a new Table 1 that summarizes this information.

A	B	C	D	E	F
Co-assembly	Number of BGCs	Full-length BGCs	No. sequenced Previously	Sequencing Platforms	Metagenomes used for co-assembly/assembly
2 Flye_co-assembly	300	67	28	PacBio Sequel (n=4)	m54017_180413_173154, m54017_180414_134655, m54017_180414_220614, m54017_180417_205359
3 Combined_Sequel_Illumina	459	31	42	PacBio Sequel (n=4) & Illumina HiSeq 2500 (n=2)	m54017_180413_173154, m54017_180414_134655, m54017_180414_220614, m54017_180417_205359, 11774.4.218925, 12041.5.235284
4 Ultimate_Sequel+Sequell	1191	419	21	PacBio Sequel (n=4) & PacBio Sequel II (n=1)	m54017_180413_173154, m54017_180414_134655, m54017_180414_220614, m54017_180417_205359, pbio-2210.20021
5 pbio-1768.15750	16	0	3	PacBio RS II	pbio-1768.15750
5 pbio-1772.15782	5	0	1	PacBio RS II	pbio-1772.15782
7 pbio-1768.15751	0	0	0	PacBio RS II	pbio-1768.15751
3 m54017_180413_173154	24	3	8	PacBio Sequel	m54017_180413_173154
3 m54017_180414_134655	30	2	7	PacBio Sequel	m54017_180414_134655
0 m54017_180414_220614	0	0	0	PacBio Sequel	m54017_180414_220614
1 m54017_180417_205359	44	7	5	PacBio Sequel	m54017_180417_205359
2 pbio-2210.20021	548	174	9	PacBio Sequel II	pbio-2210.20021
3 11774.4.218925	139	3	19	Illumina HiSeq 2500	11774.4.218925
4 12041.5.235284	232	6	32	Illumina HiSeq 2500	12041.5.235284

Ln 117: “This co-assembly was as large as our hybrid co-assembly of two short-read Illumina libraries and the four Sequel long-read libraries produced with metaSPAdes”

3. (Ln 141) *The used NCBI database is old. How would the results change if you used a new one?*

This is a valid point. Clearly using a more contemporary database could yield more matches for any sequencing-based study as more of the natural environment is sequenced and made publicly available. It is likely that we would have more matches in 2021 than when we ended our analysis. That said, there have not been many new biocrust metagenomes deposited onto the NCBI since

our search (although new biocrust soil metagenomes from the Chihuahuan Desert were deposited on July 19 2021 [<https://www.ncbi.nlm.nih.gov/bioproject/748083>]).

4. (Ln 171) *Please add parameters of bbmap use.*

We have added the following sentence:

Ln 195: “To calculate secondary metabolite gene transcription after wetting we mapped the individual read transcripts to each contig containing a BGC using BBMap (ref=assembled metagenome, in=filtered metatranscriptomic sequences, outm=transcripts mapped to contigs.sam)”.

5. (Ln 174) *Is there a motivation for the cutoff at five mapped reads per gene? The number 5 seems pretty arbitrary.*

The number of 5 reads per gene was selected to exclude very low levels of mapping (i.e. singletons or doubletons) that would amplify mapping rates, but was sensitive enough to allow for low-level changes in mapping rate to be observed. This has been revised:

Ln 199: “Remarkably, we found that 395 genes from 240 BGCs were transcribed at all timepoints (using a threshold of at least five mapped transcripts per gene within a cluster at a single time point which excluded low levels of read mapping)”.

6. (Ln 175, Ln 189, Fig. 2b) *Labeling of the y-axis in figure 2b suggests that it indicates which percentage of all RNA reads mapped to genes of the respective category. However, the text in the aforementioned Lns specifies the percentage of genes which show expression that clears a certain threshold. Please clarify what is shown in the figure and adjust the labeling if necessary.*

Good point and thank you for catching this mistake. We have changed the graph to reflect that it is the proportion, not percentage, of genes are transcribed at each timepoint on the y-axis.

The updated figure caption for Fig. 2B reads as follows: “Core bacterial gene transcription ($n=46$ genes including DNA-binding proteins, Large and Small subunit ribosomal proteins) shown in blue compared to secondary metabolite gene transcription (orange). Genes transcribed at all 10 timepoints (rightmost point) are thought to experience constitutive expression. The y-axis indicates the proportion of genes transcribed at each timepoint category.”

The edited figure is shown here:

7. (Ln 182) Please clarify: Is this observation of other genes based on mapping against BCG-containing contigs only?

No, it's against all genes across all contigs. We have added the following text.

Ln 223: "Of the 966,111 'non-secondary' genes detected across all contigs, i.e. those with or without BGCs"

8. (Ln 187) Spelling/grammar. "[...] focused on 46 core metabolic [...]"

Thank you for spotting this error. We have changed it as you have suggested.

Ln 226 to: "We then focused on 46 core metabolic bacterial genes".

9. (Ln 201) Is this an adjusted *p* value (multiple test correction)?

Thank you for pointing this out! This was an oversight on our part and re-analysis to correct for multiple testing did not support our original assertion. When we redid the analysis we found that *Bacteroidetes* were also positively correlated which is very interesting as we describe below. We also performed additional analysis looking at the number of BGCs, also provided below. Therefore we have revised the text to the following:

Ln 245: "This analysis revealed some segregation of biosynthetic gene transcription by taxonomy including for the *Cyanobacteria* and a few other phyla (Fig. 3a). A two-sided Pearson pairwise correlation analysis revealed that *Bacteroidetes* were the phyla most strongly correlated with the cyanobacteria (Pearson's $R = 0.858$, adjusted $p < 0.004$, with a one-step Bonferroni correction, Table S8). This is interesting because *Bacteroidetes* have recently been identified as one of the

keystone members of the biocrust cyanosphere (Couradeau, Giraldo-Silva et al. 2019) and are also a common host-associated organism (Magne, Gotteland et al. 2020).”

10. (Ln 207) *Please explain further the correlation that was measured here. What data served as input?*

Thank you for pointing this out. We have provided a detailed description as shown below:

Ln 400: “Pearson two-sided pairwise correlations were calculated on normalized values using Pingouin (v 0.4.0) Python package with a one-step Bonferroni correction. Here, the average Z-score of mapped transcripts to contigs with BGCs were compared among phyla using the aforementioned filter of at least 20 mapped transcripts. The correlations consider the average Z-score pattern across all timepoints (Table S8) after which Benjamini-Hochberg FDR correction is applied to the p-value statistic.”

11. (Ln 220) *Which test produced this p value?*

This was a Pearson correlation using an analysis of covariance (ANOCOVA) with the Benjamini-Hochberg FDR correction applied, revised text below:

Ln 245: “To further examine the phylogenetic conservation of BGC transcription among phyla we analyzed a subset of biosynthetic genes individually ($n=12,470$ genes) using t-SNE visualization (Van Der Maaten 2014)”

12. (Ln 247 to 257) *The Materials and Methods section does not clearly explain whether there are biological differences in the samples that were used in the various sequencing experiments. Please elucidate: Are all sequencing experiments based on the same biological sample? If yes, please address the different results regarding taxonomic composition (shown in Fig. 1b). If not, please indicate which sequencing library is associated with which biological sample.*

This is a great question, and we are sorry for causing confusion. Yes, all sequencing was performed on the same DNA sample isolated from the same biocrust material. We believe that the differences in taxonomic composition stem from differences in the recovery of full-length 16S rRNA genes from the assemblies.

13. (Ln 332) *What purpose does the metatranscriptome assembly serve? Supplement S5 suggests that a search for BGCs was carried out in the MT-assembly, but this is not described, only mentioned briefly in Ln 54 of the supplement text. Table S2 suggests the existence of several*

different metatranscriptome assemblies for different time points, but again, this is not mentioned in the text.

We have added text to this part of our analysis to the paper. The purpose was to determine if we could recover whole 'operons' of BGC transcription. We have added the following sentences to link this data in the main text on

Ln 217: "Furthermore, we recovered 88 BGCs from our assembled metatranscriptomes, 59 of which were transcribed 11.5 hours after wetting. Most BGCs found in this dataset were NRPSs while cyanobacteria were the most common source of these full-transcribed BGCs (Table S4, Supplementary Results)."

14. (Ln 333) *Suggest replacing "transcripts" with "reads" for clarity.*

Thanks, we have made this change on:

Ln 386: "The unassembled reads were then mapped".

15. (Ln 333) *It is my understanding that RNA reads were only mapped to metagenome contigs containing biosynthetic gene clusters. Using a complete assembly for the mapping step would keep reads from erroneously being mapped to BCGs if the mapper can find a better alignment on a non-BCG region. Please elaborate on the reason for choosing only the BCG-contigs as targets. Please provide details regarding the parameters that bbmap was used with.*

Thank you for catching this issue, we apologies for the oversight. The mapping was performed on all contigs > 5kb, whether or not they contained a BGC. Here is an example script that we ran on a fasta file containing all contigs:

```
module load python
```

```
./bbmap.sh ref=Sequel_Sequelll_co-assembly.fasta
```

```
./bbmap.sh          in=Biocrust_Metatranscriptome_15min_B_prinseq_good_Fhe4.fastq
```

```
outm=Sequel_Sequelll_co-assembly_15min_B.sam
```

We revised the text as such:

Ln 382 to the following: "The unassembled reads were then mapped to assembled metagenomic contigs using bbmap v38.73."

16. (Ln 335) *To facilitate reproducibility, please elaborate on the use of DESeq2 (design formula and any non-default parameters used like type of differential expression test or type of multiple comparison adjustment).*

We only used default parameters in the DESeq2 portion of the experiment. The updated text reads as follows on:

Ln 390: “Next, we used DESeq2 v1.28.0 (Love, Huber et al. 2014) under default parameters in the R statistical environment v3.6.3 to test which genes underwent differential expression by explicitly testing expression against the control sample (0 hours)”

17. (Ln 339) *Please provide details regarding this step. What is meant by transcripts mapping to 16S Data?*

Here we meant the transcripts that mapped to all identified full-length 16S rRNA genes that we extracted earlier. These would indicate taxa that are actively transcribing at each time point. We have updated the text to show this on:

Ln 411: “We also mapped transcripts to the 16S rRNA gene data to estimate microbial transcription for each phylum.”

18. (Ln 389) *BioProject accession is invalid, correct accession is PRJNA691698.*

Thanks for catching this mistake, we have changed the accession appropriately on:

Ln 452: PRJNA691698.

19. *Please provide a citation regarding the metatranscriptome sequencing. The paper by Rajeev et al. (Reference 40) provides details about the biological samples, but does not mention RNA-sequencing.*

Thank you for spotting this omission. The total RNA was prepared in Rajeev *et al.*, 2013 the sequencing was performed in the paper “Isolation of a significant fraction of non-phototroph diversity from a desert Biological Soil Crust” by Nunes da Rocha *et al.*, 2015.

We have added this reference to the text (Ln 373) and it is the new Reference 47.

20. (Figure 2c) *Predicted structures are not shown on the right as the description of the figure would indicate. The description lacks an explanation of what exactly is shown between the heatmaps. Most importantly, it is unclear what the individual rows in the heatmap represent.*

Assuming each row represents one of the 13 time points, what is the base transcription level used to calculate fold changes? Or is the heatmap displaying transformed count data?

This is another good point. Our original caption was incomplete. We completely updated the caption for Figure 2C so that it now reads as follows: “**c**, Two putative rearranged siderophore-producing gene clusters found in the co-assembled metagenomes that showed homology are aligned. The heatmaps indicate the transcription of each gene in the BGC based on statistical tests against the control sample; 0 hours at the point of wetting. Heatmap columns are scaled to the size of the mapped gene, and row order ($n=13$ rows) indicates progression across the experiment from 0 hours (bottom row) to 3 days after wetting (top row). The left-hand column shows day (yellow) and night (purple) categories. The predicted chemical structure of NODE_86 is shown above while the structure of NODE_81 is below the contig. Transcriptional profiles of gene clusters with differentially expressed genes at night are shown with stars.”

As explained in the methods, we used DESeq2 to test for significant differences in transcription for each genes compared to 0 hours (the control). The heatmaps illustrate these differences. The text on Ln 386 reads as follows: “Next, we used DESeq2 v1.28.0 (Love, Huber et al. 2014) under default parameters in the R statistical environment v3.6.3 to test which genes underwent differential expression by explicitly testing expression against the control sample (0 hours). Here we tested two environmental treatments, (i) the diurnal cycling regime (i.e., day to night to day) and, independently, (ii) the influence of wetting and drying.

21. Tables S4 and S6 are delivered in a PDF format which makes them very difficult to read. Please provide these as xlsx files like the other tables in the supplement.

We apologize for this issue. We have uploaded all tables as .xlsx files now.

22. (Figure S7) It is unclear what is shown here (e.g. fold changes, p-values). Please provide additional details in the description of the figure and add a label to the color chart.

These are genes that showed Log2 fold changes compared to the control (0 hours after wetting).

The updated text reads as follows: “**Fig. S7. | DESeq2 showed significantly enriched gene transcription at night.** Genes labelled on the heatmaps were those located within BGCs, while unlabeled rows are ‘non-BGC’ genes. Heatmap colors are based on DESeq2 comparisons to the control (0 hours – bottom row) based on Log2Fold changes. All genes are significantly differentially transcribed at night. Left color axis indicates the condition, i.e. day (yellow) or night (purple). **a**, Flye co-assembly genes. **b**, Ultimate co-assembly genes.” on

Ln 117 of the Supplemental Information.

23. (Figure S8) Lacks labeling of the color charts.

Thanks for spotting this. We have added the following text to the caption;

“Fig. S8. | Transposase transcription. Heatmap showing the transcriptional Log₂Fold change of all transposases located in BGCs over time, with 0 hours the bottom row and 3 days after wetting the top row as in Fig. S7. We identified 3 broad categories of expression: (i) unexpressed (no mapped transcripts), (ii) weak- to moderate-expression, and (iii) strongly expressed transposases. Reds indicate higher levels of transcription while blues are lowly-transcribed.”

24. Sometimes, the language seems a bit ultra-exaggerating (“ultra-long contigs”, “ultra-large assemblies”, “ultra-long gene clusters”, etc.)

Reviewer 1 had the same sentiment. We have simply switched to “long contigs” in the revised manuscript on Lns 75, 83, 121 and 136.

References

- Bowker, M. A., S. C. Reed, F. T. Maestre and D. J. Eldridge (2018). *Biocrusts: the living skin of the earth*, Springer.
- Büdel, B., T. Dulić, T. Darienko, N. Rybalka and T. Friedl (2016). Cyanobacteria and algae of biological soil crusts. *Biological soil crusts: an organizing principle in drylands*, Springer: 55-80.
- Bushnell, B. (2014). BBMap: a fast, accurate, splice-aware aligner, Lawrence Berkeley National Lab.(LBNL), Berkeley, CA (United States).
- Couradeau, E., A. Giraldo-Silva, F. De Martini and F. Garcia-Pichel (2019). "Spatial segregation of the biological soil crust microbiome around its foundational cyanobacterium, *Microcoleus vaginatus*, and the formation of a nitrogen-fixing cyanosphere." *Microbiome* **7**(1): 1-12.
- Fernandes, V. M., N. M. Machado de Lima, D. Roush, J. Rudgers, S. L. Collins and F. Garcia-Pichel (2018). "Exposure to predicted precipitation patterns decreases population size and alters community structure of cyanobacteria in biological soil crusts from the Chihuahuan Desert." *Environmental microbiology* **20**(1): 259-269.
- Frank, J. A., Y. Pan, A. Tooming-Klunderud, V. G. Eijsink, A. C. McHardy, A. J. Nederbragt and P. B. Pope (2016). "Improved metagenome assemblies and taxonomic binning using long-read circular consensus sequence data." *Scientific reports* **6**(1): 1-10.
- Giraldo-Silva, A., C. Nelson, N. N. Barger and F. Garcia-Pichel (2019). "Nursing biocrusts: isolation, cultivation, and fitness test of indigenous cyanobacteria." *Restoration Ecology* **27**(4): 793-803.
- Hiraoka, S., Y. Okazaki, M. Anda, A. Toyoda, S.-i. Nakano and W. Iwasaki (2019). "Metaepigenomic analysis reveals the unexplored diversity of DNA methylation in an environmental prokaryotic community." *Nature communications* **10**(1): 1-10.
- Howe, A. C., J. K. Jansson, S. A. Malfatti, S. G. Tringe, J. M. Tiedje and C. T. Brown (2014). "Tackling soil diversity with the assembly of large, complex metagenomes." *Proceedings of the National Academy of Sciences* **111**(13): 4904-4909.
- Karaoz, U., E. Couradeau, U. N. da Rocha, H.-C. Lim, T. Northen, F. Garcia-Pichel and E. L. Brodie (2018). "Large blooms of Bacillales (Firmicutes) underlie the response to wetting of cyanobacterial biocrusts at various stages of maturity." *MBio* **9**(2): e01366-01316.
- Kearse, M., R. Moir, A. Wilson, S. Stones-Havas, M. Cheung, S. Sturrock, S. Buxton, A. Cooper, S. Markowitz and C. Duran (2012). "Geneious Basic: an integrated and extendable desktop software platform for the organization and analysis of sequence data." *Bioinformatics* **28**(12): 1647-1649.
- Kidron, G. J. and S. Y. Tal (2012). "The effect of biocrusts on evaporation from sand dunes in the Negev Desert." *Geoderma* **179**: 104-112.
- Kupriyanova, E. V., M. A. Sinetova, A. G. Markelova, S. I. Allakhverdiev, D. A. Los and N. A. Pronina (2011). "Extracellular β -class carbonic anhydrase of the alkaliphilic cyanobacterium *Microcoleus chthonoplastes*." *Journal of Photochemistry and Photobiology B: Biology* **103**(1): 78-86.
- Li, H., B. Handsaker, A. Wysoker, T. Fennell, J. Ruan, N. Homer, G. Marth, G. Abecasis and R. Durbin (2009). "The sequence alignment/map format and SAMtools." *Bioinformatics* **25**(16): 2078-2079.
- Love, M. I., W. Huber and S. Anders (2014). "Moderated estimation of fold change and dispersion for RNA-seq data with DESeq2." *Genome biology* **15**(12): 550.
- Lynch, M. D. and J. D. Neufeld (2015). "Ecology and exploration of the rare biosphere." *Nature Reviews Microbiology* **13**(4): 217-229.
- Magne, F., M. Gotteland, L. Gauthier, A. Zazueta, S. Poesa, P. Navarrete and R. Balamurugan (2020). "The firmicutes/bacteroidetes ratio: a relevant marker of gut dysbiosis in obese patients?" *Nutrients* **12**(5): 1474.

Nayfach, S., A. P. Camargo, F. Schulz, E. Eloë-Fadrosh, S. Roux and N. C. Kyrpides (2021). "CheckV assesses the quality and completeness of metagenome-assembled viral genomes." Nature biotechnology **39**(5): 578-585.

Van Der Maaten, L. (2014). "Accelerating t-SNE using tree-based algorithms." The Journal of Machine Learning Research **15**(1): 3221-3245.

REVIEWERS' COMMENTS:

Reviewer #1 (Remarks to the Author):

The manuscript has improved enormously and is now largely acceptable for publication. I have one suggestion to improve conceptually. The name BCGs (Microbial biosynthetic gene clusters) is rather vague and I wonder if the authors might do an effort to improve this denomination. In any case, secondary metabolites are always coded by the flexible genome (as opposed to the core) and this fact together with the pangenomic perspective of the topic has been nearly completely skipped. I understand that this is irrelevant for the biotechnological perspective but very critical for the Genomics-Microbiology side. I can recommend the reading of a manuscript that we have had as a preprint for some time (long-read metagenomics, the next step?) and now published in *Frontiers* as "Enhanced recovery of Microbial Genes and Genomes From a Marine Water Column Using Long-Read Metagenomics. Since I am one of the authors (Rodriguez-Valera) I cannot compel the use of this citation (although the bioarxiv preprint predates their first submission to *Commsbiol*).

Reviewer #2 (Remarks to the Author):

I would like to thank the authors for addressing most of my comments and making the paper much more clearer while adding more details. I do not have further comments, except for one. Caption for Table 1 is missing.

Response to Reviewers Comments

REVIEWERS' COMMENTS:

Reviewer #1 (Remarks to the Author):

*The manuscript has improved enormously and is now largely acceptable for publication. I have one suggestion to improve conceptually. The name BCGs (Microbial biosynthetic gene clusters) is rather vague and I wonder if the authors might do an effort to improve this denomination. In any case, secondary metabolites are always coded by the flexible genome (as opposed to the core) and this fact together with the pangenomic perspective of the topic has been nearly completely skipped. I understand that this is irrelevant for the biotechnological perspective but very critical for the Genomics-Microbiology side. I can recommend the reading of a manuscript that we have had as a preprint for some time (long-read metagenomics, the next step?) and now published in *Frontiers* as "Enhanced recovery of Microbial Genes and Genomes From a Marine Water Column Using Long-Read Metagenomics. Since I am one of the authors (Rodriguez-Valera) I cannot compel the use of this citation (although the bioarxiv preprint predates their first submission to *Commsbiol*).*

We appreciate the positive feedback on our revised manuscript. We also agree with this important point. We have added the following sentence to include the value of this reference: "Previous work has demonstrated the potential for deep shotgun metagenomic sequencing to directly characterize BGCs from environmental samples, but the assembly of full-length BGCs from short reads is associated with significant major limitations. Notably, BGCs are almost always part of the flexible, rather than core, genome, which can assemble poorly using short read metagenomes (Haro-Moreno et al., 2021)."

The usage of the term biosynthetic gene cluster (BGC), however, is entirely consistent with the current accepted nomenclature for these genetic elements, please refer to Medema *et al.*, 2015, "Minimum Information about a Biosynthetic Gene cluster" in *Nature Chemical Biology*.

Reviewer #2 (Remarks to the Author):

I would like to thank the authors for addressing most of my comments and making the paper much more clearer while adding more details. I do not have further comments, except for one. Caption for Table 1 is missing.

Thank you for your original comments, the paper is certainly better for the constructive feedback. We have added a caption for Table 1. It reads as follows: “**Table 1.** The number of biosynthetic gene clusters recovered from each assembly or co-assembly are shown with details regarding full-length BGCs, BGCs sequenced previously and which assemblies contributed to the co-assemblies.”